# Resolution as a Direction: Vector-Panning Feature Alignment for Cross-Resolution Re-Identification

Zanwu Liu [* 1]   Chao Yuan [* 2]   Bo Li [1]   Xiaowei Zhang [3]   Guanglin Niu [1]

## Abstract

Cross-resolution person re-identification (CR-ReID) remains challenging in practical surveillance, where camera quality and capture distance lead to substantial resolution gaps between low-resolution (LR) queries and high-resolution (HR) gallery images. Prior approaches commonly rely on super-resolution (SR) or resolution-invariant representation learning, which often increases system complexity and may not directly address the feature mismatch induced by resolution degradation. In this work, we report a new empirical finding from a dedicated analysis in which identity-specific variation is averaged out: the HR–LR feature discrepancy produced by standard ReID backbones exhibits a consistent, resolution-related semantic direction in the embedding space. We further support this observation with statistical analyses based on Canonical Correlation Analysis (CCA) and Pearson correlation analysis. Motivated by this finding, we propose Vector Panning Feature Alignment (VPFA), a lightweight post-hoc module that learns to pan LR features along the learned resolution direction to obtain pseudo-HR representations. VPFA operates after feature extraction and can be integrated into existing ReID systems with negligible overhead. Extensive experiments on multiple CR-ReID benchmarks show that VPFA achieves state-of-the-art performance while improving efficiency compared to SR-based or jointly trained alternatives. Code is available at https://github.com/ashmentlzw/VPFA

---
[*]Equal contribution  [1]School of Artificial Intelligence, Beihang University, Beijing, China [2]School of Computer Science and Engineering, Beihang University, Beijing, China [3]College of Computer Science and Technology, Qingdao University, Qingdao, China. Correspondence to: Guanglin Niu <beihangngl@buaa.edu.cn>.

*Proceedings of the 43$^{rd}$ International Conference on Machine Learning*, Seoul, South Korea. PMLR 306, 2026. Copyright 2026 by the author(s).

## 1. Introduction

Person re-identification (ReID) aims to match images of the same person captured by different cameras (Ye et al., 2022a). Thanks to deep learning advances (Dosovitskiy et al., 2020; Wang et al., 2022), ReID has achieved strong performance in controlled settings. However, in real-world scenarios, many factors would degrade the performance such as viewpoint, lighting, and especially resolution gaps (Ren & Zhang, 2024; Zheng et al., 2022; Wu et al., 2023; Ye et al., 2022b). Specifically, the image resolution usually differs due to varying quality and distances of cameras. Thus, directly applying standard ReID models in this scenario leads to notable performance drops. To tackle this, many CR-ReID methods use super-resolution (SR) to enhance LR inputs or design resolution-invariant representations (Han et al., 2020; Zhang et al., 2021; Li et al., 2015; Wu et al., 2023). However, these models face some challenges: Super resolution (SR) only enhances visual details and improves apparent resolution, but is not designed specifically for person ReID tasks. Therefore, as shown in Fig.1's Challenge, the performance of SR-based models heavily depends on the results of SR while its capability of cross resolution feature alignment for ReID is limited. In contrast, resolution invariant approaches must align cross resolution features while preserving identity discrimination, where identity differences and resolution differences are entangled in the feature space, causing difficulty in training an effective model that could accurately align cross-resolution pedestrian features. To address these challenges, we rethink CR-ReID task with a more straightforward perspective. Rather than altering image resolution or enforcing cross-resolution alignment during training, we attempt to exploit a two-stage feature alignment strategy. In the first stage, we disentangle resolution-specific discrepancies from identity-discriminative features by explicitly modeling cross-resolution differences. In the second stage, we align features at comparable resolutions to ensure identity consistency.

Our idea stems from a simple but insightful observation. Mikolov (Mikolov et al., 2013) showed that semantic relations can be represented as consistent vector offsets. As shown in Fig.1's Inspiration, for instance, Vec(King) − Vec(Man) + Vec(Woman) ≈ Vec(Queen), indicating that

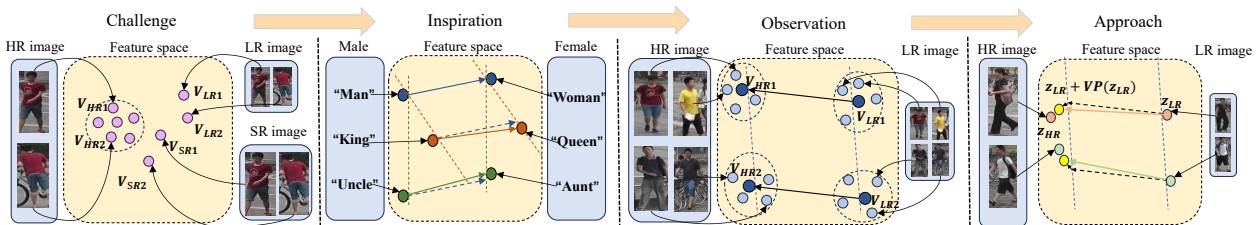

*Figure 1.* The motivation of our paper. Top: semantic offset in word embedding space. Mid: resolution direction in feature space. Bottom: our Vector Panning-based cross-resolution Feature Alignment (VPFA).

*Table 1.* Cosine similarity between the average HR–LR difference vectors computed from two disjoint ID subsets on the Market-1501 and CUHK03 datasets. Higher similarity indicates more consistent resolution-specific offsets across identities.

| Dataset | 2× | 3× | 4× |
|---------|------|------|------|
| Market-1501 | 0.9933 | 0.9914 | 0.9935 |
| CUHK03 | 0.9960 | 0.9967 | 0.9977 |

the gender difference between words corresponds to a vector with stable direction and magnitude, allowing similar semantic relations to be modeled through comparable offsets. While such offsets are not strictly linear, they effectively capture meaningful semantic variations at the feature level. Furthermore, DeepMind's recent work (Neel et al.) shows that multilayer perceptrons (MLP) can store factual associations and perform retrieval retrieval via a direct feature transformation strategy. **Inspired by these significant observations, we attempt to explore whether the feature space of CR-ReID also contain analogous directions that encode resolution differences**.

To investigate this, as shown in Fig.1's Observation, we conduct an empirical analysis on Market-1501 with 2× downsampling. We extract features from both the original and downsampled images to form $S_{ori}$ and $S_{2x}$, and further split each set into two halves to obtain $S_{ori1}$, $S_{ori2}$, $S_{2x1}$, and $S_{2x2}$. For each subset, we compute identity-wise average features and then average across identities to derive four representative vectors: $V_{HR1}$, $V_{HR2}$, $V_{LR1}$, and $V_{LR2}$. Finally, we measure the cosine similarity between the difference vectors $V_{HR1} - V_{LR1}$ and $V_{HR2} - V_{LR2}$. As shown in Table 1, the consistently high similarity values suggest that resolution-induced feature shifts converge to a stable semantic direction once identity information is averaged out. **This finding provides strong empirical evidence that resolution differences in CR-ReID are not random noise but structured semantic variations in the feature space**. Moreover, Section Section 3 provides a more rigorous statistical justification, reinforcing the significance of this resolution-specific semantic direction. These results highlight the existence of a resolution-aware semantic direction, which motivates us to explicitly model and exploit this direction for more robust cross-resolution feature alignment. This averaging procedure is used only for semantic analysis and is not involved in VP training.

Based on our finding, we identify a new approach to CR-ReID. The fundamental challenge in CR-ReID is that identity information and resolution discrepancy are coupled in the features, which interferes with identity discrimination. Since we have demonstrated the existence of a resolution semantic direction in the feature space, we directly model this resolution discrepancy and translate the original LR feature vector to mitigate the impact of resolution differences. Therefore, as illustrated in Fig.1's Approach, we propose **Vector Panning based Cross resolution Feature Alignment (VPFA)**. We design an MLP based Vector Panning (VP) module to learn the semantic offset between HR and LR features. We also introduce a Vector Panning Loss (VPL) to align both direction and magnitude. Our method operates after feature extraction and achieves feature alignment between HR and LR images at very low cost. Our contributions are summarized as follows:

- We introduce a new perspective for CR-ReID and experimentally discover and validate a semantic direction of resolution in the feature space.

- Based on this finding, we design a lightweight postprocessing module that models resolution differences in features, namely VPFA.

- Extensive experiments on four benchmarks show that VPFA outperforms existing CR-ReID models and offers insights for broader cross-modality tasks.

## 2. Related Work

**Person Re-ID** Person Re-Identification has seen rapid progress in both algorithm design and performance over the past decade (Bai et al., 2019; Li et al., 2020; 2019; Chen et al., 2018; Shen et al., 2018; Martinel et al., 2019; Zhao et al., 2017; Yuan et al., 2025c). Many methods focus on handling challenges from background clutter, pose variation, and occlusion (Wu et al., 2021; Yuan et al., 2025b; Pang et al., 2024; Dou et al., 2023; Wu et al., 2019; Fang et al., 2023; Liu et al., 2022; Li et al., 2018; Liu et al., 2018). To capture fine-grained local cues, part-based models are widely adopted, explicitly or implicitly modeling body structures (Zhou et al., 2020; Wu et al., 2022; Wei et al., 2021;

2022). Moreover, significant efforts aim to reduce annotation reliance through unsupervised learning (Chen et al., 2021; Yin et al., 2023; Pang et al., 2024), domain adaptation (Hu et al., 2022; Lee et al., 2023; Pang et al., 2022), and weak supervision (Zhu et al., 2019; Jin et al., 2023). Despite these advances, resolution variation remains insufficiently addressed, limiting the robustness and practicality of ReID.

*Table 2.* Statistical analysis results for cross-resolution feature relationships. (a) CCA results comparing LR–HR feature matrices with random matrices. (b) Pearson correlation statistics across different resolution gaps.

*(a)* CCA results between LR–HR feature matrices and random matrices on Market-1501

| Scale | Cross-res (R1/R2/R3) | Random (R1/R2/R3) |
|---|---|---|
| 2× | 0.4826 / 0.4186 / 0.4725 | 0.2482 / 0.3358 / 0.2117 |
| 3× | 0.4944 / 0.4948 / 0.4840 | 0.3007 / 0.3407 / 0.3366 |
| 4× | 0.4907 / 0.3479 / 0.4759 | 0.3054 / 0.2945 / 0.2780 |

*(b)* Grouped Pearson Correlation Statistics for Different Resolution Gaps

| Metric | 2× | 3× | 4× |
|---|---|---|---|
| $r$ | 0.5373 | 0.5533 | 0.5721 |
| Std. of $r$ | 0.0803 | 0.0748 | 0.0699 |
| Proportion of $r > 0.4$ | 94.54% | 96.93% | 98.53% |

**Cross-Resolution Person ReID** Current CR-ReID approaches fall into two categories: learning resolution-invariant representations (Li et al., 2015; Wang et al., 2016; Wu et al., 2023) and super-resolution (SR) based methods (Jiao et al., 2018; Wang et al., 2018; Cheng et al., 2020; Zhang et al., 2021; Chen et al., 2019; Li et al., 2019; Wu et al., 2023). For example, Li et al. (Li et al., 2015) combine cross-scale alignment with multi-scale metric learning, and Wu et al. (Wu et al., 2023) introduce resolution-adaptive masks to disentangle resolution-specific information. However, most such methods emphasize identity-level features, neglecting the broader HR–LR feature relationship. SR-based methods restore image details before recognition. SING (Jiao et al., 2018) jointly trains SR and ReID modules, while CSR-Net (Wang et al., 2018) stacks SRGANs (Ledig et al., 2017) for gradual refinement. INTACT (Cheng et al., 2020) adds compatibility constraints between SR and ReID branches, and PS-HRNet (Zhang et al., 2021) integrates channel attention and HRNet in a pseudo-siamese design. Beyond visual enhancement, RAIN (Chen et al., 2019) and CAD-Net (Li et al., 2019) employ adversarial learning to enforce resolution invariance. LRAR (Wu et al., 2023) further decomposes HR features into shared and unique components to preserve HR-specific detail. However, due to the inherent domain gap between HR and LR, SR-based methods still struggle to reach standard ReID performance. In contrast,

we model the global HR–LR shift directly in embedding space and use a lightweight postprocessing module to convert LR features into pseudo-HR ones, enabling effective cross-resolution alignment without SR or retraining.

## 3. Theoretical Analysis

### 3.1. Theoretical Inspiration

As the example shown in Fig.1: $\text{Vec}(\text{King}) - \text{Vec}(\text{Man}) + \text{Vec}(\text{Woman}) \approx \text{Vec}(\text{Queen})$, our idea is inspired by the linear semantic structures observed in word embedding spaces. To verify the effectiveness of this idea in the feature space of CR-ReID, we aim to demonstrate that there exists a directional discrepancy between features of high resolution and low resolution images, and we regard this directional transformation as a "resolution vector" analogous to semantic shifts in word embeddings.

### 3.2. Statistical Justification

To verify the existence of a shared semantic direction induced by resolution differences, we perform global and local analyses on the Market-1501 dataset.

**Canonical Correlation Analysis** (CCA) evaluates global linear relationships between HR and LR features. It finds canonical vectors that maximize inter-set correlation. As shown in Table 2a, canonical correlations between LR and HR feature matrices are significantly higher than those from random feature pairs. Based on standard thresholds (Johnson & Wichern, 1988), values above 0.4–0.5 indicate a moderate-to-strong linear association, strongly evidencing of semantic alignment between LR and HR representations.

**Pearson Correlation Analysis** assesses local consistency across individuals. A global "resolution shift" vector is computed by averaging HR–LR feature differences. For each of 50 identities, individual difference vectors are grouped into 25 pairs, whose mean vectors are then correlated with the global shift. In the absence of semantic alignment, these correlations would cluster around zero. However, Table 2b shows most groups exceed 0.5 correlation, considered moderate to strong per Cohen (Cohen, 1988). Furthermore, correlations increase with larger resolution gaps, indicating stronger alignment.

These CCA and Pearson correlation results jointly provide strong evidence for the existence of a consistent semantic direction induced by resolution variation, observable at both global and identity-specific levels.

## 4. Method

Based on our finding, we propose a novel postprocessing method for CR-ReID, namely Vector Panning based Feature

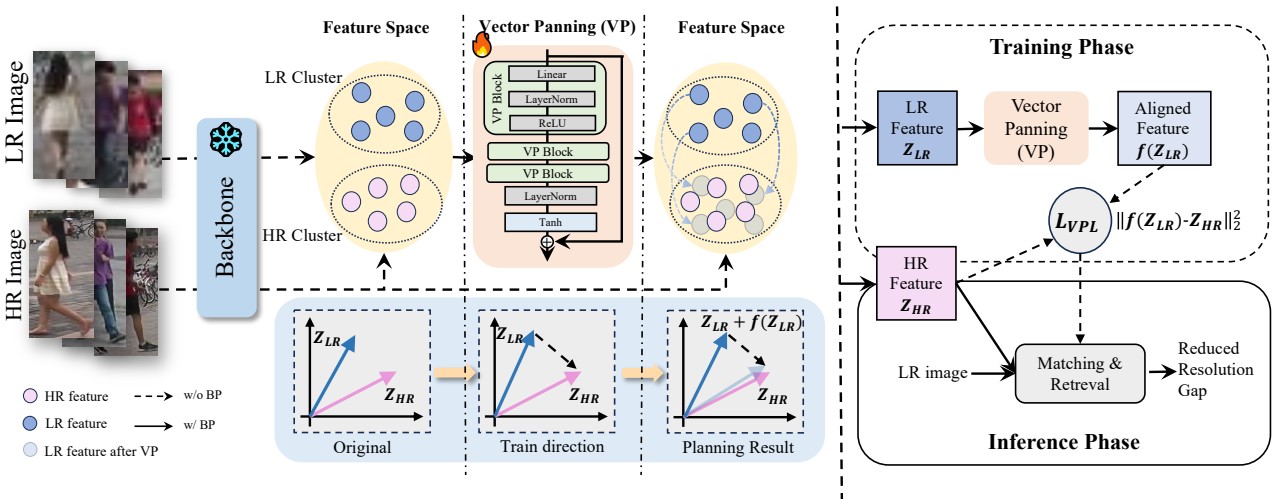

*Figure 2.* An overview of the proposed Vector Panning based Feature Alignment (VPFA) framework.

Alignment (VPFA). The core concept is to model the discrepancy between HR and LR features in the feature space so as to minimize the impact of resolution on features. In the experiment suggesting that, at the aggregate level, resolution variation lies in a low-rank feature subspace with a consistent dominant semantic direction, the features are extracted by standard ReID models with publicly available weights; therefore, our method also does not require training the ReID model. As shown in Fig.2, VPFA first uses a frozen ReID model to extract features from LR and HR images. Initially, LR and HR features are separated in the embedding space. The VP module, composed of stacked VP blocks (Linear + LayerNorm + ReLU), learns a semantic shift that aligns LR features with HR clusters. A final Tanh gate and a residual connection produce pseudo HR features. The vector diagram illustrates how VP models and compensates for resolution induced directional differences.

### 4.1. Problem Formulation

Let $z_{LR}, z_{HR} \in \mathbb{R}^d$ denote the backbone extracted features from the LR and HR images of the same person, respectively. Our objective is to learn a transformation function $g : \mathbb{R}^d \to \mathbb{R}^d$, implemented via a lightweight module, such that:

$$\hat{z}_{LR} = g(z_{LR}) \approx z_{HR} \qquad (1)$$

where $\hat{z}_{LR}$ represents the pseudo-HR feature vector aligned by the VP Module. The mapping $g$ is designed to eliminate the semantic distortions introduced by resolution degradation while preserving identity discriminative information.

### 4.2. Vector Panning (VP) Module Architecture

The core of our VPFA framework is the Vector Panning (VP) Module, which learns a transformation $f(\cdot)$ that maps LR features $z_{LR}$ into pseudo HR representations $\hat{z}_{LR}$. The

transformation is implemented as a gated residual function:

$$\hat{z}_{LR} = z_{LR} + \text{Gate}(\text{VP}(z_{LR})) \qquad (2)$$

in which **VP (Vector Panning)** refers to a three-layer non-linear transformation, where each layer consists of a linear projection followed by Layer Normalization and ReLU activation. It is compactly expressed as:

$$\text{VP}(z_{LR}) = W_3 \, \sigma_2(W_2 \, \sigma_1(W_1 z_{LR})) \qquad (3)$$

The output of VP is passed through a Tanh activation, denoted as **Gate**, which regulates the magnitude of the residual vector:

$$f(z_{LR}) = \tanh(W_4 \cdot \text{VP}(z_{LR})) \qquad (4)$$

**Residual design.** Instead of predicting $z_{HR}$ directly, the residual structure allows the model to learn an offset that selectively corrects the resolution-specific distortions in $z_{LR}$. This design preserves identity semantics encoded by the backbone while simplifying the learning objective. It also makes optimization easier and prevents overfitting in low data regimes.

**Expressiveness of VP.** The VP block serves as a high-capacity nonlinear mapping that captures the semantic transformation between LR and HR domains. The layered composition progressively adjusts the embedding along resolution-sensitive subspace directions, which is particularly effective in modeling feature variations caused by resolution degradation.

**Stabilization with LayerNorm.** Each transformation inside VP is followed by LayerNorm, which normalizes feature statistics per instance. This mitigates internal covariate shifts and stabilizes training. As resolution-induced variance leads to heteroscedastic feature distributions, LayerNorm is

critical for learning transferable mappings across resolution domains.

**Controlled correction via Gate.** The final Gate layer applies a $\tanh$ activation to the residual vector. This constrains each dimension to $[-1, 1]$, acting as a soft clipping mechanism. It ensures that the correction remains within a semantically meaningful range and prevents feature explosion. Additionally, the symmetric range supports both additive and subtractive corrections, which is essential for precise vector alignment.

To ensure training stability and prevent early overfitting, we initialize all weight matrices using a zero-mean Gaussian distribution $\mathcal{N}(0, 10^{-3})$ and set all biases to zero. This leads to $f(z_{LR}) \approx 0$ at the beginning of the training, which means that the initial output satisfies $\hat{z}_{LR} \approx z_{LR}$. Such an initialization strategy is inspired by residual tuning techniques in large scale vision models, where starting from an identity mapping allows the model to retain the backbone's semantic structure and incrementally learn useful corrections. In our setting, this prevents VP from prematurely altering discriminative identity information in $z_{LR}$ and encourages gradual, resolution aware adjustment over time.

### 4.3. Vector Panning Loss (VPL)

The proposed Vector Panning Loss (VPL) supervises the VP using the Mean Squared Error (MSE) between the predicted pseudo-HR feature $\hat{z}_{LR}$ and the target HR feature $z_{HR}$. Formally, we define:

$$\mathcal{L}_{\text{VPL}} = \|\hat{z}_{LR} - z_{HR}\|_2^2 \qquad (5)$$

Although VPL is mathematically identical to MSE, it is specifically designed to guide the learning of resolution-aware residual transformations in the feature space.

Let $\theta$ denote the angle between $\hat{z}_{LR}$ and $z_{HR}$, and $r = \|\hat{z}_{LR}\|$, $R = \|z_{HR}\|$ as their respective norms. Expanding the squared Euclidean distance using the law of cosines:

$$\mathcal{L}_{\text{VPL}} = r^2 + R^2 - 2rR\cos\theta \qquad (6)$$

This shows that minimizing $\mathcal{L}_{\text{VPL}}$ aligns both the direction via $\cos\theta$ and magnitude of the features. Since most person ReID systems adopt cosine similarity as the inference metric, this dual effect makes VPL naturally compatible with downstream matching.

In contrast to contrastive or triplet loss functions, VPL does not require explicitly mining positive or negative pairs. Since our goal is not to classify or separate identities but to correct resolution-induced distortions, we directly align paired LR and HR features at the sample level. This enables stable, efficient training without relying on hard pair mining.

---

**Algorithm 1** Proposed Model Training

---

**Input:** Training image pairs $\{(I_{LR}, I_{HR})\}$; pretrained backbone $B(\cdot)$
**Output:** Trained Vector Panning Module $f(\cdot)$
**Stage 1: Feature Extraction**
(1) For each image pair $(I_{LR}^i, I_{HR}^i)$, extract paired features: $z_{LR}^i = B(I_{LR}^i)$, $z_{HR}^i = B(I_{HR}^i)$.
**Stage 2: Training the VP Module**
(1) Initialize the module $f(\cdot)$ with Gaussian weights;
(2) For $epoch = 1$ to $T$ do
    (a) Sample a mini-batch of paired features $\{(z_{LR}^i, z_{HR}^i)\}$;
    (b) Compute predictions: $\hat{z}_{LR}^i = f(z_{LR}^i)$;
    (c) Compute loss $\mathcal{L}_{\text{VPL}}$ using Eq. (2);
    (d) Update $f(\cdot)$ via backpropagation.
**end for**

---

### 4.4. Model Training

Unlike existing cross-resolution ReID methods that rely on adversarial training or joint end-to-end pipelines, our approach decouples feature correction from backbone learning and requires no modification to the base ReID model. This enables our framework to serve as a lightweight plug-in that enhances cross-resolution robustness in a post-hoc manner.

We assume that a standard ReID model has been trained on HR images. Given paired LR and HR images, we extract their features using the frozen backbone and directly construct paired feature samples $\{(z_{LR}, z_{HR})\}$. Our training is conducted in a fully pairwise manner, where each LR feature is directly aligned with its corresponding HR counterpart. No feature aggregation or identity-level averaging is required.

The VP module is trained to regress the HR feature from its paired LR feature by minimizing the proposed Vector Panning Loss (VPL):

$$\mathcal{L}_{\text{VPL}} = \|f(z_{LR}) - z_{HR}\|_2^2. \qquad (7)$$

The VP is optimized using the Adam optimizer with a fixed learning rate schedule and weight decay, and the model typically converges within 120 epochs. The process is simple, efficient, and supervision-free. The full training pipeline is summarized in Algorithm 1.

## 5. Experiment

In this section, we first introduce the datasets, experimental settings, and implementation details. We then conduct extensive experiments, including comparisons with existing methods and ablation studies, to validate the effectiveness of our approach. Additionally, we evaluate the extensibility

of our method by applying it to two representative models in cross-modal ReID and text-image ReID, in order to examine its potential generalization ability for cross-domain feature alignment.

## 5.1. Datasets

We used four person ReID benchmarks in our CR-ReID evaluations. The **Market-1501** dataset (Zheng et al., 2015) consists of 32,668 images of 1,501 identities captured in 6 camera views. We used the standard 751/750 training/test identity split. The **CUHK03** dataset (Gray & Tao, 2008) contains 14,097 images of 1,467 individuals captured in 10 (5 pairs) different cameras. Besides, we use the 1,367/100 training/testing identity split (Li et al., 2019). The **VIPeR** dataset (Li et al., 2014) contains 1,220 images of 72 person identities captured by 2 cameras. We randomly divided this dataset into two non-overlapping halves based on the identity labels. The **CAVIAR** dataset (Zheng et al., 2017) is a dataset collected in the real world. The dataset contains 1,220 images of 72 individuals captured from two cameras. Following (Jiao et al., 2018), we discard 22 individuals who have only HR images. The remaining data is randomly divided into two equal parts for training and testing, each containing 25 identities. Following (Li et al., 2019), we evaluate our method under the multiple LR (MLR) person ReID setting. We conduct experiments on four synthetic benchmarks: Market-1501, CUHK08, VIPeR, and one real-world dataset, CAVIAR. For the synthetic datasets, query images from one camera are downsampled by a randomly selected rate $r \in \{2, 3, 4\}$, resulting in image sizes of $H/r \times W/r$, where $H$ and $W$ are the original height and width. The gallery images remain at the original resolution. We refer to these modified datasets as MLR-Market, MLR-CUHK08, MLR-VIPeR, and MLR-DukeMTMC. In contrast, CAVIAR provides naturally captured images with real resolution differences, making it a more realistic MLR benchmark.

## 5.2. Experimental Environment

All experiments were conducted on a server equipped with an Intel(R) Xeon(R) Platinum 8336C CPU (base frequency 2.30 GHz) as the central processing unit. For parallel computing and acceleration of deep learning tasks, the system utilized NVIDIA A800 GPUs with 40GB of memory. The operating system running on the server was Debian GNU/Linux 11(bullseye), providing a stable environment for software execution and dependency management.

## 5.3. Implementation Details

We adopt TransReID (He et al., 2021) as the backbone for feature extraction, trained only on original-resolution datasets (official weights used when available). No fine-tuning is performed for cross-resolution settings. During

inference, both HR and LR images are fed into the backbone to obtain features $z_{HR}$ and $z_{LR}$. Our VP module is a three-layer MLP with residual connection. It first projects the 3840-dimensional(-d) input to 2048-d, followed by two hidden layers of size 2048, each with a linear projection, LayerNorm, and ReLU. The final layer maps back to 3840-d with a Tanh activation to constrain the residual. The output is added to the input to produce the aligned feature $\hat{z}_{LR}$. To build training data, we directly use paired sample-level features extracted from HR and LR image pairs, i.e., $(z_{LR}, z_{HR})$, without identity-level averaging. The averaging operations described in our semantic-analysis experiments are only for analysis and are not used to train VP. We sample 5,000 feature pairs for training with random shuffling. The VP is trained for 120 epochs using Adam (lr = 2e-4, weight decay = 1e-5) with batch size 32. The loss is mean squared error (VPL) between $\hat{z}_{LR}$ and $z_{HR}$. We initialize linear layers with $\mathcal{N}(0, 10^{-3})$ and set all biases to zero. At inference, the trained VP is directly applied to LR features as a postprocessing step.

## 5.4. Comparisons to State-of-the-Art Methods

From the perspective of backbone architecture, the methods in Table 3 cover the mainstream families used in current visual recognition. The ResNet-based group includes ResNet-50 (He et al., 2016), CamStyle (Zhong et al., 2018), FD-GAN (Ge et al., 2018), CSR-GAN (Wang et al., 2018), CAD (Li et al., 2019), JBIM (Zheng et al., 2022), LRAR (Wu et al., 2023), and RAPSR (Yuan et al., 2025a), where ResNet-50 or residual encoders are used as the main ReID feature extractor. The non-ResNet CNN-based group includes Part Aligned (Zhao et al., 2017), PyrNet (Martinel et al., 2019), SING (Jiao et al., 2018), RAIN (Chen et al., 2019), PRI (Han et al., 2020), PyrNet+PRI (Han et al., 2020), INTACT (Cheng et al., 2020), and PS-HRNet (Zhang et al., 2021), which are built on convolutional part/pyramid networks, SR/GAN pipelines, or HRNet-style high-resolution CNNs. The Transformer/ViT-based group includes IFRSW (Peng et al., 2024), which adopts Swin Transformer/SwinIR components, and our VPFA, which uses TransReID (He et al., 2021) with a ViT backbone. Therefore, our comparison spans ResNet-style CNNs, other CNN architectures, and ViT/Transformer architectures, covering the backbone families commonly adopted in existing visual recognition tasks.

We compare our proposed VPFA with two categories of representative methods. (1) Traditional person ReID baselines, including ResNet-50, CamStyle, and FD-GAN, which are not specifically designed for resolution variation. (2) Recent advanced CR-ReID methods that aim to address cross-resolution challenges via super-resolution, resolution-invariant learning, or hybrid strategies. Representative examples include SING (Jiao et al., 2018), PS-HRNet (Zhang

*Table 3.* Results of Cross-Resolution Re-ID (%). Bold and underlined numbers indicate top two results, respectively.

| method | MLR-Market-1501 | | | MLR-CUHK03 | | | MLR-VIPeR | | | CAVIAR | | |
|---|---|---|---|---|---|---|---|---|---|---|---|---|
| | Rank1 | Rank5 | Rank10 | Rank1 | Rank5 | Rank10 | Rank1 | Rank5 | Rank10 | Rank1 | Rank5 | Rank10 |
| ResNet-50 (He et al., 2016) | 57.0 | 78.7 | - | 70.8 | 91.3 | - | 31.4 | 63.1 | - | 31.1 | 65.5 | - |
| CamStyle (Zhong et al., 2018) | 74.5 | 88.6 | 93.0 | 69.1 | 89.6 | 93.9 | 34.4 | 56.8 | 66.6 | 32.1 | 72.3 | 85.9 |
| Part Aligned (Zhao et al., 2017) | 75.6 | 88.5 | 92.2 | 73.4 | 92.1 | 97.5 | 40.2 | 62.3 | 73.1 | 35.7 | 71.4 | 87.9 |
| FD-GAN (Ge et al., 2018) | 79.6 | 91.6 | 93.5 | 73.4 | 93.8 | 97.9 | 39.1 | 62.1 | 72.5 | 33.5 | 71.4 | 96.5 |
| PyrNet (Martinel et al., 2019) | 83.8 | 93.3 | 95.6 | 83.9 | 97.1 | 98.5 | - | - | - | 43.6 | 79.2 | 90.4 |
| SING (Jiao et al., 2018) | 74.4 | 87.8 | 91.6 | 67.7 | 90.7 | 94.7 | 33.5 | 57 | 66.5 | 33.5 | 72.7 | 89 |
| CSR-GAN (Wang et al., 2018) | 76.4 | 88.5 | 91.9 | 71.3 | 92.1 | 97.4 | 37.2 | 62.3 | 71.6 | 34.7 | 72.5 | 87.4 |
| RAIN (Chen et al., 2019) | - | - | - | 78.9 | 97.3 | 98.7 | 42.5 | 68.3 | 79.6 | 42.0 | 77.3 | 89.6 |
| RAPSR (Yuan et al., 2025a) | - | - | - | 74.9 | 94.6 | - | - | - | - | 42.8 | 78.4 | - |
| CAD (Li et al., 2019) | 83.7 | 92.7 | 95.8 | 82.1 | 97.4 | 98.8 | 43.1 | 68.2 | 77.5 | 42.8 | 76.2 | 91.5 |
| PRI (Han et al., 2020) | 84.9 | 93.5 | 96.1 | 85.2 | 97.5 | 98.8 | - | - | - | 43.2 | 78.5 | 91.9 |
| PyrNet+PRI (Han et al., 2020) | 86.9 | 93.8 | 96.4 | 86.5 | 97.7 | 99.1 | - | - | - | 45.2 | 84.1 | 94.6 |
| INTACT (Cheng et al., 2020) | 88.1 | 95.0 | 96.9 | 86.4 | 97.4 | 98.5 | 46.2 | 73.1 | 81.6 | 44.0 | 81.8 | 93.9 |
| JBIM (Zheng et al., 2022) | 88.1 | 95.1 | 96.9 | 88.3 | 97.2 | 98.7 | 49.7 | 72.5 | 81.3 | 52.0 | 83.1 | 94.4 |
| LRAR (Wu et al., 2023) | 90.1 | 96.2 | 97.7 | 89.2 | _98.9_ | _99.8_ | - | - | - | _63.6_ | 79.2 | _96.6_ |
| PS-HRNet (Zhang et al., 2021) | 91.5 | 96.7 | _97.9_ | 92.6 | 98.3 | 99.4 | 48.7 | 73.4 | 81.7 | 48.2 | 84.5 | 96.3 |
| IFRSW (Peng et al., 2024) | _92.3_ | _96.9_ | 97.8 | _93.4_ | 98.7 | 99.2 | _50.1_ | _75.3_ | _83.4_ | 52.4 | _87.3_ | **97.9** |
| ours | **94.1** | **97.6** | **98.6** | **94.7** | **99.5** | **99.8** | **50.5** | **78.8** | **87.6** | **66.6** | **87.9** | 93.6 |

et al., 2021), and INTACT (Cheng et al., 2020), among others. The comparison results are summarized in Table 3, and the following observations can be made:

(1) Our **VPFA** achieves the highest Rank-1 accuracy on all four datasets. It outperforms PS-HRNet by 2.6% and 2.1% on MLR-Market-1501 and MLR-CUHK03 respectively, setting new state-of-the-art results. It also ranks first in Rank-5 and Rank-10 on three of the four datasets.

(2) Compared to SR-based CSR-GAN and hybrid INTACT, VPFA shows clear advantages. On MLR-Market-1501, it improves Rank-1 by 17.7% over CSR-GAN and 6.0% over INTACT, demonstrating the effectiveness of feature-level alignment over pixel-level restoration or joint learning.

(3) On smaller datasets, VPFA remains competitive. It reaches 50.6% Rank-1 on MLR-VIPeR, outperforming JBIM (49.7%) and PS-HRNet (48.7%). On CAVIAR, it achieves 66.6% Rank-1 and 87.9% Rank-5, exceeding the previous bests of 63.6% and 84.5%.

(4) Against resolution-invariant methods like LRAR and PS-HRNet, VPFA achieves comparable or better results. This confirms the effectiveness of our lightweight post-processing in addressing resolution gaps without architectural changes or retraining.

## 5.5. Efficiency Analysis

We assess the efficiency of our VPFA module from two key perspectives: inference speed and model size. On average, our method achieves an inference efficiency of 4,424,371.31 samples/second, demonstrating its extremely low computational overhead. Additionally, VPFA contains only 24.14 million parameters, which is significantly smaller than many

existing ReID modules. In our environment, the baseline TransReID model runs at 80.80 samples/second, while the standalone VPFA module reaches 4.4M samples/second, indicating that the overall inference speed is still dominated by the backbone and the additional cost introduced by VPFA is negligible. These results underscore the efficiency and practicality of our approach. As a lightweight, feature-level module, VPFA can be seamlessly integrated into existing ReID frameworks without retraining, making it highly suitable for rapid deployment in real world scenarios.

## 5.6. Ablation Study and Parameter Analysis

*Table 4.* Ablation analysis and parameter analysis of VPFA on MLR-Market1501. Bold values indicate the best results. "Dim" refers to the hidden dimension in the VP block.

*(a)* Baseline and residual analysis

| Category | Method / Setting | Rank-1 | Rank-5 | Rank-10 |
|---|---|---|---|---|
| Baseline | TransReID | 90.3 | 96.1 | 97.5 |
| | TransReID + VPFA | **94.1** | **97.6** | **98.6** |
| Residual | w/o Residual | 91.2 | 96.7 | 98.0 |
| | w/ Residual | **94.1** | **97.6** | **98.6** |

*(b)* Dimension and VP blocks analysis

| Category | Method / Setting | Rank-1 | Rank-5 | Rank-10 |
|---|---|---|---|---|
| Dim | 512 | 92.7 | 96.9 | 97.9 |
| | 1048 | 93.4 | 97.2 | 98.5 |
| | 2048 | **94.1** | **97.6** | **98.6** |
| | 3840 | 92.8 | 96.9 | 98.0 |
| #VP Blocks | 1 | 92.8 | 97.0 | 98.0 |
| | 2 | 93.2 | 97.6 | 98.1 |
| | 3 | **94.1** | **97.6** | **98.6** |
| | 4 | 92.6 | 96.9 | 97.8 |

**With vs. Without VPFA** Unless otherwise noted, all ablations are conducted on MLR-Market-1501.To assess the impact of VPFA, we compare the baseline TransReID model with and without it, as shown in Table 4a. VPFA consistently improves all metrics, notably boosting Rank-1 by +3.8%, without altering the backbone or its training.

**Effect of Residual Connection** We ablate the residual connection by replacing the output with a direct mapping $f(z_{LR})$. As shown in Table 4a, its removal degrades performance, confirming its necessity. The residual design enables VP to refine features without overwriting identity semantics, aiding robustness to resolution variation.

**Effect of hidden layer dimension** We vary the hidden layer width of the VP structure, as shown under Dim. VP Block in Table 4b. Performance improves to 2048 before saturation, with 2048-d hidden layers achieving 94.1% Rank-1 on MLR-Market1501. Smaller widths like 512 underfit the cross-resolution alignment transformation, while larger ones such as 3840 add parameters without benefits. Thus, we use 2048.

**VPblock numbers** To assess depth, we stack VP blocks. Table 4b shows increasing the number of blocks from 1 to 3 improves steadily, while adding a fourth block slightly degrades accuracy, likely due to overfitting or redundancy. Thus, moderate stacking boosts representation capacity while keeping the design concise.

### 5.7. Visualization Analysis

**Feature Space Visualization** To further verify the effectiveness of our method in aligning LR features with their HR counterparts, we visualize feature distributions with t-SNE before and after VPFA. In Fig.3, colors denote identities and brightness encodes resolution, with dark for HR and light for LR.From Fig.3(a), LR features are dispersed from their HR counterparts with weak intra-identity cohesion, revealing resolution-induced divergence. In contrast, Fig.3(b) shows that after applying VPFA, the LR features shift to HR features and form tighter identity clusters. These results show that VPFA semantically aligns cross-resolution features and reduces the domain gap.

### 5.8. Interpretability Analysis

To further understand what VPFA modifies in the feature space, we visualize similarity-based Grad-CAM maps for low-resolution query images, as shown in Fig.4. Concretely, for each LR query and its matched HR gallery image, we take the cosine similarity between their global features as a scalar score, and backpropagate its gradient to the patch embedding layer of the ViT backbone. We then follow the standard Grad-CAM procedure to aggregate gradients

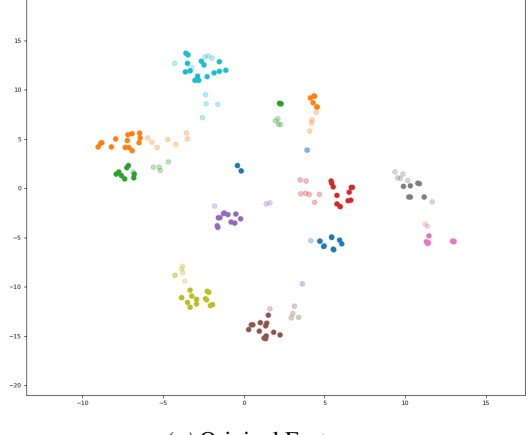

*(a)* Original Features

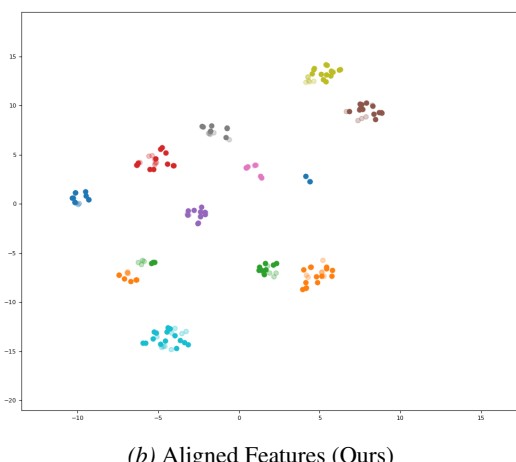

*(b)* Aligned Features (Ours)

*Figure 3.* t-SNE visualization of features from 12 identities. Colors indicate different person IDs, and color brightness reflects resolution level (dark: HR, light: LR). VPFA effectively narrows the gap between HR and LR representations.

over channels, upsample the resulting importance map to the image resolution, and overlay it as a heatmap on the original image. The brighter (red) regions therefore indicate spatial locations that contribute more to increasing the LR–HR matching score. Comparing the baseline and VPFA-enhanced maps, we observe that VPFA suppresses many scattered, resolution-induced responses on the background and low-informative regions, while consistently strengthening the focus on the human body and fine-grained clothing textures. This suggests that VPFA effectively reduces the noise introduced by resolution discrepancies and encourages the model to rely more on semantically meaningful, identity-related details for cross-resolution matching.

## 6. Conclusion

We propose Vector Panning–based Feature Alignment (VPFA), a post-processing method for cross-resolution person ReID. Unlike prior approaches that require super-resolution or backbone changes, VPFA works purely at the

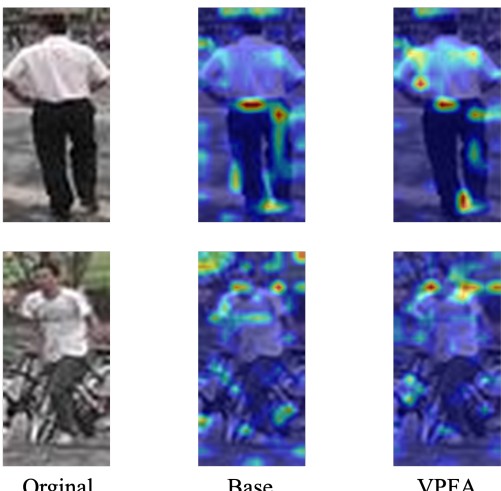

|          |        |      |
|:--------:|:------:|:----:|
| Orginal  | Base   | VPFA |

*Figure 4.* Similarity-based Grad-CAM visualization before and after VPFA on low-resolution queries.

feature level and integrates into existing ReID models without retraining. Central to VPFA is a lightweight MLP-based Vector Panning (VP) module that learns correction vectors to align low-resolution features with high-resolution ones. The Vector Panning Loss (VPL) enforces both directional and magnitude consistency. Experiments on four benchmarks demonstrate that VPFA consistently improves strong baselines and generalizes to cross-modality ReID, underscoring its versatility as a universal feature alignment module.

## 7. limitations

Although VPFA shows promising performance as a lightweight post-hoc feature-alignment module, its scope has several limitations.

First, VPFA relies on the frozen ReID backbone to extract meaningful identity-discriminative features. When the input resolution is extremely low or identity cues are severely lost, feature-space panning cannot fully recover missing visual evidence, and a clear gap to standard high-resolution ReID may remain.

Second, our analyses suggest that the HR–LR discrepancy is structured rather than random, but they do not prove the existence of a single universal one-dimensional direction. A more cautious interpretation is that resolution-induced variation may lie in a structured, possibly low-rank and input-dependent subspace, which motivates the use of an MLP-based nonlinear mapping.

Third, VPFA is trained with image-level LR–HR pairs and is mainly evaluated under standard CR-ReID protocols based on synthetic downsampling, with additional validation on real low-resolution data. Its robustness to more complex real-world degradations, such as motion blur, compression artifacts, sensor noise, non-uniform local resolution changes,

and camera-dependent resolution gaps, remains underexplored.

Finally, since VPFA operates on final embedding features, it aims to reduce resolution-induced feature bias rather than reconstruct truly missing high-frequency or spatial details. Therefore, it should be viewed as complementary to super-resolution or feature-reconstruction methods rather than a replacement for them.

## Impact Statement

This paper presents work whose goal is to advance the field of Computer Vision. There are many potential societal consequences of our work, none which we feel must be specifically highlighted here.

## Acknowledgements

This work is partially supported by the National Natural Science Foundation of China (No. U25A20531, No. 62376016).

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

# A. Appendix

## A.1. Testing in More Challenging Cross-resolution Settings

To further validate the generalizability and robustness of our approach, we conduct experiments under more complex cross-resolution scenarios. Specifically, we follow the traditional cross-resolution ReID construction strategy for consistency, and extend the original MLR datasets to include six downsampling ratios (2×, 3×, 4×, 5×, 6×, and 7×), forming a new benchmark we refer to as **MLRpro**. In addition, we evaluate on MSMT17, which is the largest fully supervised ReID dataset, comprising 126,441 images with greater domain and resolution diversity.

As shown in Table 5, our proposed VPFA consistently improves the performance over both the standard MLR and the more challenging MLRpro settings. The improvements are especially significant on MSMT17, demonstrating that VP maintains strong effectiveness even under large-scale, high-variance conditions.

*Table 5.* Performance under more challenging cross-resolution settings.

| Method | Market1501 (R1/5/10) | MSMT17 (R1/5/10) |
|---|---|---|
| MLR | 90.3 / 96.1 / 97.5 | 80.4 / 88.7 / 91.4 |
| MLR + VP | **94.1 / 97.6 / 98.6** | **83.1 / 90.5 / 91.9** |
| MLRpro | 84.3 / 90.4 / 92.2 | 75.1 / 83.9 / 87.1 |
| MLRpro + VP | **88.2 / 93.7 / 95.6** | **79.6 / 86.2 / 90.0** |

## A.2. Performance under Separate Downsampling Ratios

Following the suggestion to report results under separate downsampling settings, we conduct additional experiments with fixed resolution gaps of $2\times$, $3\times$, and $4\times$ downsampling, respectively. As shown in Table 7, VPFA consistently improves retrieval performance across all ratios, demonstrating robust effectiveness under varying degrees of resolution discrepancy. Notably, the improvement becomes more pronounced as the resolution gap increases, indicating that VPFA is particularly beneficial for more challenging cross-resolution scenarios.

## A.3. Generalization across Modalities

To assess VPFA's generalization beyond cross-resolution tasks, we apply it to two cross-modality settings: visible-infrared (VI) and text-image (TI) ReID. We use SAAI (Fang et al., 2023) as the baseline on RegDB for VI ReID, and IRRA (Jiang & Ye, 2023) on RSTPReid for TI ReID. VPFA is directly integrated into each backbone without architectural or training changes.

**RegDB** (Nguyen et al., 2017) contains 8,240 visible/infrared image pairs from 412 identities, split into train/test sets. It includes two settings: Visible2Infrared (v2i) and Infrared2Visible (i2v), referring to cross-modal retrieval directions. **RSTPReid** (Zhu et al., 2021) includes 20,505 images of 4,101 identities, each annotated with two text descriptions. Following the official split, the training/validation/test sets include 3,701, 200, and 200 identities, respectively.

As shown in Table 6, VPFA consistently improves performance across both tasks. On RegDB, it enhances SAAI by +0.68% and +1.01% mAP in the v2i and i2v settings, respectively, with corresponding Rank-1 gains of +0.29% and +0.62%. These improvements verify VPFA's effectiveness in mitigating modality-induced shifts between visible and infrared features.

On the more challenging TI task, VPFA boosts IRRA on RSTPReid by +0.68% mAP, +1.00% Rank-1, and +0.75% Rank-5. This suggests its capability in aligning representations even across vision and language domains.

Overall, these results highlight VPFA's strong cross-modal generalization. Its feature-level semantic correction mechanism

*Table 6.* Generalization of VPFA to cross-modality and text-image ReID tasks. VI: visible-infrared; TI: text-image.

| Task | Method | Dataset | mAP | Rank-1 | Rank-5 |
|---|---|---|---|---|---|
| VI | SAAI (v2i) | RegDB | 91.45 | 91.07 | - |
| | +VPFA (v2i) | | **92.13** | **91.36** | - |
| | SAAI (i2v) | | 92.01 | 92.09 | - |
| | +VPFA (i2v) | | **93.02** | **92.71** | - |
| TI | IRRA | RSTPReid | 47.17 | 60.20 | 81.30 |
| | +VPFA | | **47.85** | **61.20** | **82.05** |

*Table 7.* Results under separate $2\times$, $3\times$, and $4\times$ downsampling settings (Rank-1/5/10, %).

| Scale | VPFA (r1/r5/r10) | Original (r1/r5/r10) |
|-------|------------------|----------------------|
| $2\times$ | 94.6 / 97.8 / 99.0 | 94.3 / 98.2 / 98.9 |
| $3\times$ | 94.0 / 97.4 / 98.3 | 91.1 / 96.8 / 97.9 |
| $4\times$ | 93.6 / 97.2 / 98.0 | 85.6 / 93.3 / 95.1 |

*Table 8.* Supplementary results across different ReID frameworks and backbone families on MLR-Market-1501 (Rank-1/5/10, %).

| Model | VPFA (r1/r5/r10) | Original (r1/r5/r10) |
|-------|------------------|----------------------|
| TransReID (ViT) | 94.1 / 97.6 / 98.6 | 90.3 / 96.1 / 97.5 |
| TransReID (DeiT) | 91.7 / 96.5 / 97.7 | 89.1 / 95.7 / 97.1 |
| CLIP-ReID (ViT) | 93.4 / 97.3 / 98.6 | 90.5 / 96.0 / 97.2 |
| CLIP-ReID (CNN) | 90.6 / 96.1 / 97.4 | 88.2 / 95.6 / 97.1 |

enables flexible deployment across modalities without retraining, making it a broadly applicable postprocessing solution for heterogeneous ReID tasks.

### A.4. Generalizability across Different Backbones

We further examine the generalizability of VPFA across different backbone architectures and training frameworks. In the **cross-modal** ReID setting, we adopt the SAAI framework with a **ResNet**-based backbone, which differs from the **Transformer**-based backbone used in our main cross-resolution experiments. Under this distinct architectural choice and experimental setting, VPFA still produces consistent improvements, suggesting that the proposed feature-alignment module is not tightly coupled to a single backbone family or ReID framework.

To provide a more direct validation in the **cross-resolution** ReID setting, we conduct additional experiments on MLR-Market-1501 with multiple ReID frameworks and backbone families. Specifically, we evaluate VPFA on **TransReID** with both **ViT** and **DeiT** backbones, and on **CLIP-ReID** with both **ViT** and **CNN** backbones. As shown in Table 8, VPFA consistently improves the corresponding original baseline across all four settings. For example, VPFA improves Rank-1 accuracy from 90.3% to 94.1% on TransReID (ViT), from 89.1% to 91.7% on TransReID (DeiT), from 90.5% to 93.4% on CLIP-ReID (ViT), and from 88.2% to 90.6% on CLIP-ReID (CNN). The CLIP-ReID (CNN) result is particularly useful because it indicates that the benefit of VPFA is not limited to Transformer-based feature extractors.

These results should be interpreted as within-backbone and within-framework comparisons, where each VPFA variant is compared against its corresponding original baseline under the same feature extractor and evaluation protocol. Together with the ResNet-based cross-modal results, they support the practical generalizability of VPFA across different feature extractors, while we do not claim that VPFA universally dominates all methods under every possible matched training setting.

### A.5. Comparison with General Post-hoc Feature Adaptation Methods

To further validate the effectiveness of VPFA and situate it within the broader literature on post-hoc feature adaptation, we compare VPFA with several representative alternatives on MLR-Market1501. In our ablations, the residual connection in VPFA is removed by default, where the resulting non-residual MLP already serves as a commonly adopted baseline for feature alignment. Beyond this, we additionally consider two widely used adaptation modules: **FiLM** (Perez et al., 2018), a general conditioning layer frequently used in visual reasoning, and **Adapter** (Houlsby et al., 2019), a parameter-efficient tuning module popularized in NLP and later adopted in multimodal settings. Moreover, we include a simple **mean vector offset** baseline as a lightweight post-hoc correction strategy.

For a fair comparison, we replace the VPFA head with each alternative using standard configurations, while keeping the training protocol and loss function unchanged. As summarized in Table 9, VPFA consistently achieves the best Rank-1/5/10 performance among all evaluated methods, demonstrating its clear advantage for post-hoc feature correction in cross-resolution person re-identification.

### A.6. Feature Alignment Effectiveness

To further validate the effectiveness of VPFA in aligning cross-resolution features, we conduct a quantitative analysis on the MLR-Market1501 dataset. Specifically, we compute the Euclidean distances between the HR (HR) and LR (LR) feature centers for a subset of representative identities, both before and after applying VPFA.

*Table 9.* Comparison with general post-hoc feature adaptation alternatives on MLR-Market1501 (Rank-1/5/10, %).

| Method | Rank-1 / Rank-5 / Rank-10 |
|---|---|
| VPFA | 94.1 / 97.6 / 98.6 |
| VPFA (no residual) | 91.2 / 96.7 / 98.0 |
| FiLM (Perez et al., 2018) | 92.1 / 97.2 / 98.4 |
| Adapter (Houlsby et al., 2019) | 91.8 / 97.4 / 98.4 |
| Offset (mean vector) | 91.2 / 96.5 / 97.7 |

*Figure 5.* Visualization of Retrieval Results Before and After Applying Vector Panning based Feature Alignment(VPFA).(The left side of the figure shows the original experimental results, while the right side shows the results after applying VPFA.)

As shown in Table10, the distances between HR and LR feature centers are consistently reduced after applying our module. On average, the distances are reduced by approximately 25%, clearly demonstrating VPFA's capability in narrowing the semantic gap induced by resolution differences. These results provide strong evidence that our method effectively aligns cross-resolution features at the representation level.

*Table 10.* Euclidean distances between HR and LR feature centers for six sample identities on MLR-Market1501.

| ID | 1401 | 865 | 877 | 1355 | 507 |
|---|---|---|---|---|---|
| Before VPFA | 17.87 | 22.53 | 15.32 | 15.26 | 15.15 |
| After VPFA | 13.78 | 16.34 | 11.28 | 11.10 | 10.87 |
| Reduction | 22.92% | 27.46% | 26.35% | 27.24% | 28.18% |

## A.7. Experimental Details

For experiments on the MLR-CUHK03, MLR-VIPeR, and CAVIAR datasets, we perform 10 random splits and report the average results. The corresponding results are presented in Table11, Table12, and Table13, respectively.

## A.8. Visualization Analysis

Fig.5 presents a qualitative comparison of retrieval results before and after applying VPFA. The left side shows the original results, where many top-ranked retrievals (highlighted with red bounding boxes) are incorrect matches, indicating that the baseline model struggles to distinguish fine-grained appearance cues under cross-resolution settings. In contrast, the right side displays the results after integrating VPFA. Most top-ranked candidates are now correct matches (green bounding boxes), and the ranking of true matches has been significantly improved. This improvement demonstrates that VPFA effectively aligns LR features with their HR counterparts, reducing semantic discrepancies and enhancing retrieval precision.

## A.9. Cross-Domain Analysis

To evaluate whether our proposed VPFA module truly captures resolution-related semantics rather than dataset-specific distribution priors, we conduct cross-domain experiments where the VP is trained on one dataset and directly applied to

*Table 11.* Performance on CUHK03 over 10 splits and the average result.

| Split | 0 | 1 | 2 | 3 | 4 | 5 | 6 | 7 | 8 | 9 | AVG |
|---|---|---|---|---|---|---|---|---|---|---|---|
| R1 | 95.0 | 95.1 | 97.3 | 94.1 | 95.6 | 91.3 | 93.5 | 96.0 | 95.7 | 93.1 | 94.7 |
| R5 | 99.8 | 99.6 | 100.0 | 99.2 | 98.3 | 99.2 | 99.8 | 100.0 | 99.6 | 99.2 | 99.5 |
| R10 | 99.8 | 100.0 | 100.0 | 100.0 | 99.2 | 99.8 | 99.8 | 100.0 | 100.0 | 99.6 | 99.8 |

*Table 12.* Performance on VIPER over 10 splits and the average result.

| Split | 0 | 1 | 2 | 3 | 4 | 5 | 6 | 7 | 8 | 9 | AVG |
|---|---|---|---|---|---|---|---|---|---|---|---|
| R1 | 57.9 | 56.0 | 43.4 | 49.4 | 34.8 | 50.9 | 50.3 | 57.3 | 54.4 | 50.3 | 50.5 |
| R5 | 79.7 | 80.7 | 79.1 | 82.0 | 65.5 | 79.7 | 81.6 | 78.2 | 78.2 | 83.2 | 78.8 |
| R10 | 89.2 | 89.2 | 87.3 | 89.2 | 77.2 | 87.3 | 88.9 | 88.6 | 88.3 | 91.1 | 87.6 |

*Table 13.* Performance on CAVIAR over 10 splits and the average result.

| Split | 0 | 1 | 2 | 3 | 4 | 5 | 6 | 7 | 8 | 9 | AVG |
|---|---|---|---|---|---|---|---|---|---|---|---|
| R1 | 66.8 | 74.4 | 68.0 | 71.6 | 44.8 | 76.4 | 78.8 | 72.8 | 61.2 | 52.0 | 66.6 |
| R5 | 91.2 | 90.0 | 85.2 | 90.8 | 77.6 | 89.6 | 92.0 | 92.0 | 80.0 | 81.2 | 87.9 |
| R10 | 95.2 | 96.4 | 92.0 | 96.0 | 87.2 | 96.0 | 96.8 | 97.2 | 86.4 | 92.8 | 93.6 |

others without any fine-tuning. The results are shown in Table 14. We observe that the VP trained on MLR-Market1501 performs remarkably well on MLR-CUHK03 (Rank-1: 94.0), and vice versa (Rank-1: 90.5), even though the two datasets differ in appearance distributions and camera styles. Furthermore, both VP modules achieve considerable improvements on the CAVIAR dataset, despite no exposure to its domain during training. These results contrast with the non-cross-domain setting in Table.14, where VP is trained and tested on the same dataset. The relatively strong cross-dataset generalization suggests that VPFA does not simply overfit to dataset-specific features, but learns an intrinsic semantic direction in the embedding space that consistently reflects resolution-induced feature shifts. This demonstrates the transferable nature of the resolution alignment learned by our model.

*Table 14.* Cross-domain evaluation results of VPFA. Each row indicates the dataset used for training the VP module, while the columns show results on different test datasets.

| Training Dataset | Test Dataset | Rank-1 | Rank-5 |
|---|---|---|---|
| MLR-Market1501 | MLR-CUHK03 | 94.0 | 97.3 |
| MLR-CUHK03 | MLR-Market1501 | 90.5 | 96.2 |
| MLR-Market1501 | MLR-Market1501 | 94.1 | 97.8 |
| MLR-CUHK03 | MLR-CUHK03 | 94.7 | 99.5 |

