# OpenReview forum: "Resolution as a Direction: Vector-Panning Feature Alignment for Cross-Resolution Re-Identification"
_ICML.cc/2026/Conference — ICML 2026 regular_

### Official Review · Reviewer_Xyr7 · 2026-03-06

**Soundness:** 3
**Presentation:** 3
**Significance:** 2
**Originality:** 2
**Overall Recommendation:** 4
**Confidence:** 4

**Summary:**

The paper proposes a lightweight framework, Vector Panning (VPFA), for Cross-Resolution Person Re-identification (CR-ReID). The central claim is the discovery of a "resolution-aware semantic direction" in the feature space—a consistent vector offset between HR and LR features. The authors utilize an MLP-based "panning" module to translate LR features toward their HR counterparts, avoiding the heavy computational costs of Super-Resolution (SR) or complex resolution-invariant training.

**Compliance With Llm Reviewing Policy:**

Affirmed.

**Final Justification:**

The author ultimately addressed the issues I raised regarding FeatUp with a relatively comprehensive experiment, so I upgraded my final rating from “borderline” to “weak accept.”

**Key Questions For Authors:**

1. How does your global "panning" offset handle cases where the HR-LR difference is spatially non-uniform (e.g., a person is sharp in the torso but blurred at the feet due to motion or depth of field)?

2. Can you provide a comparison where the gallery features are upscaled using FeatUp? Does your "Vector Panning" add value when applied on top of FeatUp-restored features?

3.  If the test set is downsampled using a Gaussian kernel or real-world sensor noise instead of the Bicubic kernel used in training, does the "semantic direction" ($\Delta$) remain stable?

**Limitations:**

While the "Vector Panning" idea is elegantly simple, it feels like a step backward given the recent breakthroughs in high-resolution feature learning. By ignoring the spatial guidance and multi-scale reconstruction principles validated by FeatUp, the paper risks promoting a solution that only works on synthetic, uniformly-downsampled datasets. The authors should demonstrate that this "panning" is a superior or complementary alternative to actual feature reconstruction.

**Strengths And Weaknesses:**

**Strengths**
1. The method is significantly more efficient than standard SR-based ReID. By treating resolution alignment as a post-processing vector translation, it offers a pragmatic solution for edge-side deployment.
2. The high cosine similarity between difference vectors of disjoint identity sets is a compelling observation that suggests some degree of linear structure in resolution-induced feature shifts.

**Weaknesses**

1. The recent work FeatUp (ICLR 2024) has already established a model-agnostic framework for recovering features at any resolution. FeatUp demonstrates that "upsampling" features is not merely a global shift but requires a multi-scale, spatially-guided reconstruction using either Joint Bilateral Upsampling (JBU) or learned implicit functions.In contrast, VPFA’s "Vector Panning" assumes a global, additive offset $\vec{V}_{LR} + \Delta = \vec{V}_{HR}$. This is a significant oversimplification. While a global shift might capture the "average" degradation of a 4x downsampling, it fundamentally cannot recover the fine-grained, high-frequency details (like logos or accessory patterns) that FeatUp preserves. The authors fail to justify why a simple linear translation is sufficient when contemporary research points toward the necessity of spatially-varying refinement.

2. FeatUp is "resolution-agnostic" and works across continuous scales. VPFA, however, appears to rely on a fixed MLP trained for specific downsampling ratios (e.g., 2x, 4x). In real-world ReID, the "resolution gap" is not a discrete jump but a continuous spectrum influenced by distance and focal length. Without the downsampling-invariant properties explored in FeatUp, VPFA’s "semantic direction" may be an artifact of the specific bicubic downsampling kernel used in training rather than a robust property of the feature space.

3. The paper compares VPFA against traditional SR-based ReID methods. However, a critical baseline is missing: FeatUp-enhanced features. Given that FeatUp can upscale features from any backbone without retraining, the authors must prove that their "Vector Panning" provides any advantage over simply using a FeatUp-restored feature map. If FeatUp provides better discriminative power by recovering spatial details, the "panning" approach becomes redundant.

References
1. FeatUp: A Model-Agnostic Framework for Features at Any Resolution. ICLR 2024. Stephanie Fu*, Mark Hamilton*, Laura Brandt, Axel Feldman, Zhoutong Zhang, William T. Freeman.

---

> ### Author Rebuttal · Authors · 2026-03-31
>
> Thank you very much for your careful review. Your mention of FeatUp (ICLR 2024) is highly inspiring. As our work has mainly focused on ReID and cross-resolution ReID, we regret that we had not studied this paper in depth before. During rebuttal, we carefully read it and responded based on our current understanding; if any part is inaccurate, we would sincerely appreciate your correction. **We will also cite it in the revised manuscript and include it in the limitations discussion.**
>
> ## To Weakness 1 & Limitations:
>
> Thank you for this question. We agree that FeatUp is inspiring and shows that recovering fine spatial details often requires more than simple global operations. However, **FeatUp and VPFA address different targets, so they are not directly comparable.**
>
> FeatUp focuses on **feature-map reconstruction and upsampling to recover richer spatial details.** In contrast, VPFA operates on **the global feature before final ReID matching and aims to reduce the non-ID-related bias caused by resolution differences**, rather than recover local details such as logos or textures. Thus, FeatUp improves spatial detail, while VPFA improves feature consistency for cross-resolution matching.
>
> For cross-resolution ReID, the two methods work at different levels: **FeatUp is closer to feature-map recovery, whereas VPFA is a lightweight post-backbone correction module before matching.** They are therefore complementary rather than conflicting.
>
> In addition, VPFA is not simply a fixed linear shift. Although we use the shift form for intuition, the actual model uses an MLP with residual structure to learn an input-dependent nonlinear mapping. This is also supported by our ablation results, where directly adding a fixed shift vector performs clearly worse than the full model.
>
> Finally, VPFA is a lightweight plug-in post-processing module at inference and does not require modifying or retraining the backbone. **By contrast, methods like FeatUp, if adapted to ReID, would more likely need to be involved in backbone training or feature-map reconstruction.** Therefore, the two methods serve different purposes. We do not claim that VPFA replaces the spatial detail recovery studied by FeatUp. Rather, our point is that for matching-based tasks, especially cross-resolution ReID, reducing the non-identity-related bias caused by resolution differences is more critical than explicitly recovering all high-frequency details. We will clarify this in the revised manuscript.
>
> ## To Weakness 2:
>
> In our experiments on CUHK, **image resolutions are also not fixed, even with a fixed downsampling ratio, the actual training and inference resolutions still differ**, yet VPFA consistently improves performance. In practice, **this issue can be handled by dividing resolutions into intervals**. As for the concern that the effect may be an artifact of the bicubic kernel, our method also **performs well on the real-world cross-resolution dataset CAVIAR**, and this setting was adopted for fair comparison. Still, we acknowledge that the lack of full continuity handling is a limitation and will discuss it in the revised paper.
>
> ## To Weakness 3 & Questions 2:
>
> **Based on our understanding of FeatUp and the ReID pipeline, if FeatUp is adapted to ReID, the upsampled feature maps would still need to pass through the subsequent feature aggregation and matching head. We therefore believe that retraining or at least fine-tuning the backbone is likely needed.** In contrast, VPFA directly operates on the final matching feature and does not require backbone retraining. For comparison, we also integrated FeatUp into TransReID: we first trained FeatUp on the original ReID model, and then fine-tuned TransReID with it. The resulting Rank-1/5/10 scores were 91.2/96.6/97.3, showing only limited improvement, which also makes it closer to a super-resolution-style module. Due to time limits, we did not further tune this setting, so these results are provided for reference only.
>
> ## To Questions 1:
>
> This is a practical question, but it is beyond the scope of our current method. **Our method is designed to address inaccurate matching caused by resolution differences due to depth-of-field effects in real-world pedestrian images.** We will explore its effectiveness under such local blur conditions in future work and build a broader, more realistic cross-resolution ReID dataset for further evaluation. This is a limitation of the current work and will be discussed in the revision.
>
> ## To Questions 3:
>
> **Current cross-resolution ReID protocols and public datasets make it difficult to test under truly realistic sensor-noise conditions.** Our method is effective on CAVIAR, but this dataset is relatively small. Therefore, we cannot give a definitive answer at present. In future work, we will build a more realistic cross-resolution dataset to further evaluate our method under Gaussian blur, local blur, and other real degradations.

---

> > ### Author Rebuttal · Reviewer_Xyr7 · 2026-04-01
> >
> > It is difficult to address the Featup experiment in such a brief rebuttal; the author addressed some of my questions, and my final score was borderline.

---

> > > ### Author Response · Authors · 2026-04-03
> > >
> > > Thank you very much for your reply to our rebuttal and for your insightful suggestions. We are honored that we have already addressed some of your questions. Here, we provide further clarification on the additional points you raised, and we sincerely hope to gain your further recognition.
> > >
> > > First, **regarding your question: “Does your ‘Vector Panning’ add value when applied on top of FeatUp-restored features?”**, we have completed the corresponding experiment. Together with the previously reported FeatUp-ReID setting, the results are as follows:
> > >
> > > | MLR-Market1501              | Rank-1 | Rank-5 | Rank-10 |
> > > |----------------------------|--------|--------|---------|
> > > | TransReID                  | 90.3   | 96.1   | 97.5    |
> > > | TransReID + FeatUp         | 91.2   | 96.6   | 97.3    |
> > > | TransReID + VPFA           | 94.1   | 97.6   | 98.6    |
> > > | TransReID + FeatUp + VPFA  | 94.3   | 97.7   | 98.6    |
> > >
> > > As shown above, FeatUp can recover some fine-grained feature details and thus improves ReID performance to a certain extent. When combined with VPFA, the model benefits from both enhanced detail features and reduced feature noise caused by resolution discrepancy, leading to a more pronounced improvement in cross-resolution ReID performance.
> > >
> > > Finally, we would like to summarize the comparative strengths and limitations of VPFA and FeatUp. In our view, the key difference is that **VPFA is more specifically designed for the ReID setting**, especially for cross resolution matching in the feature space, whereas FeatUp is a more general feature recovery framework aimed at improving the spatial resolution of feature maps across broader vision tasks.
> > >
> > > Compared with FeatUp, our method has several practical advantages in the ReID framework.
> > > - **VPFA does not require retraining the backbone** and can be inserted before feature matching as a plug and play module, which makes it easy to integrate into existing ReID systems.
> > > - **VPFA is extremely lightweight**, introducing almost no additional inference cost. Its training process is also simple, requiring only paired high resolution and low resolution images in the target scenario.
> > > - Our experiments suggest that **the resolution aware semantic direction we identify exists consistently across multiple ReID tasks and frameworks**, indicating that VPFA has a certain degree of generality as a ReID oriented feature alignment module before matching.
> > >
> > > At the same time, we also acknowledge several limitations of our method compared with FeatUp, and we will discuss them explicitly in the revised paper.
> > > - **FeatUp is capable of genuinely recovering fine grained spatial details and has stronger task level generality**, whereas VPFA mainly reduces the mismatch caused by resolution discrepancy in feature space and cannot reconstruct truly missing visual details.
> > > - **FeatUp may be more beneficial for upstream tasks such as detection and tracking**, which could in turn improve downstream cross resolution ReID from a preprocessing perspective.
> > >
> > > Therefore, we do not view the two methods as strictly conflicting; rather, **FeatUp is a more general spatial feature restoration framework, while VPFA is a lightweight and more task focused solution tailored to the specific needs of ReID matching.**

---

### Official Review · Reviewer_dQAJ · 2026-03-07

**Soundness:** 4
**Presentation:** 4
**Significance:** 3
**Originality:** 4
**Overall Recommendation:** 5
**Confidence:** 3

**Summary:**

This paper proposes Vector Panning Feature Alignment (VPFA), a lightweight post-hoc framework for cross-resolution person re-identification. Identifying that resolution-induced feature discrepancies follow a consistent "semantic direction," the authors employ a gated MLP to "pan" low-resolution features into pseudo-high-resolution representations. By operating as a plug-in after feature extraction and utilizing a specialized Vector Panning Loss (VPL), VPFA eliminates the need for expensive image super-resolution or backbone retraining while achieving state-of-the-art performance.

**Compliance With Llm Reviewing Policy:**

Affirmed.

**Final Justification:**

The authors have adequately addressed my concerns and I am happy to maintain my positive rating.

**Key Questions For Authors:**

1. The experiments currently utilize TransReID (2021) as the frozen backbone. Given the rapid progress in the field, could the authors demonstrate the effectiveness of VPFA using more recent or different types of backbones, such as CLIP-ReID or other vision-language based models?
2. The heatmaps in Figure 4 do not sufficiently illustrate a clear contrast in feature focus. Could the authors provide more visualization examples in the appendix where the "panning" effect is more visually prominent, specifically showing how the attention shifts more precisely to identity-discriminative regions?
3. The submission currently lacks a dedicated discussion on the potential limitations of the VPFA framework.

**Limitations:**

Please refer to the key questions.

**Strengths And Weaknesses:**

**strengths**:

1. The paper is grounded in a strong empirical finding: resolution-induced feature discrepancies are not random but follow a consistent "semantic direction". This is supported by cosine similarity tests showing 0.99+ consistency across disjoint identity sets.
2. The paper is well-organized, moving logically from the initial observation (inspired by word embeddings) to statistical proof, methodology, and finally extensive benchmarking.
3. Applying the concept of semantic vector offsets from Natural Language Processing (NLP) to the problem of resolution degradation in Computer Vision is a creative and successful cross-disciplinary application.

**weaknesses**:

1. The effectiveness of VPFA relies on the quality of the frozen backbone (TransReID). If the backbone fails to extract any meaningful identity features from extremely low-resolution images, the "panning" may have limited utility.
2. While the empirical results are strong, the authors do not explicitly discuss the limitations of the method like the potential failure cases or the theoretical bounds of the "vector panning" assumption.

---

> ### Author Rebuttal · Authors · 2026-03-31
>
> We are truly grateful for your high evaluation of our work and for the thoughtful and constructive questions you have raised. **Your recognition and feedback are highly encouraging and very valuable to us.** In the following, we will respond to each of your questions in detail, one by one.
>
> ## To Weakness 1:
>
> Thank you for your question. Our method is a post-processing module and fundamentally relies on a conventional ReID model as the backbone. **When the image resolution is extremely low, it becomes difficult to extract effective features, and our method is indeed limited in such cases; even if some improvement is achieved, a large gap from standard ReID performance remains.** As shown in Table 5 of our paper, under extremely low downsampling settings such as 6× and 7×, our method still brings improvements, but the performance drop remains severe compared with the standard non-cross-resolution ReID setting. **This is indeed a limitation of our method, and we will add a discussion of this point in the revised manuscript.**
>
> ## To Weakness 2 & Questions 3:
>
> Thank you for your question. **We sincerely apologize that the current manuscript lacks a discussion of limitations, and we will add a detailed section on this in the revised version.** Here, we **briefly clarify the main limitations of our work.** First, the proposed method depends on the performance of the underlying standard ReID model and has limited capability when handling extremely low-resolution images. Second, in terms of generalizability, we have not yet explored whether VPFA can address other imaging degradations, such as Gaussian blur, and the current method is only designed for resolution discrepancy. Third, in terms of interpretability, we still need larger-scale and more realistic cross-resolution ReID datasets to further verify that what we capture is indeed a resolution-related semantic direction, rather than consistent noise introduced by downsampling.
>
> ## To Questions 1:
>
> Thank you for your question. **We also conducted experiments using CLIP-ReID as the backbone, and evaluated the results under different training paradigms.** In all cases, the results demonstrate the effectiveness of VPFA. The experimental results are shown in the table below.
>
> | Model | Ori (R1/R5/R10) | VPFA (R1/R5/R10) |
> |---|---:|---:|
> | TransReID (ViT)  | 90.3/96.1/97.5 | 94.1/97.6/98.6 |
> | TransReID (DeiT) | 89.1/95.7/97.1 | 91.7/96.5/97.7 |
> | CLIP-ReID (ViT)  | 90.5/96.0/97.2 | 93.4/97.3/98.6 |
> | CLIP-ReID (CNN)  | 88.2/95.6/97.1 | 90.6/96.1/97.4 |
>
> ## To Questions 2:
>
> We apologize that the current visual examples are not sufficiently intuitive. **In the revised manuscript, we will replace them with more informative examples to better illustrate the effect of our method.** We will also add larger-scale visualizations, similar to Figure 5, in the appendix of the revised version.

---

> > ### Author Rebuttal · Reviewer_dQAJ · 2026-04-02
> >
> > I would like to thank the authors for their detailed and professional response to my concerns.
> >
> > The additional experiments with CLIP-ReID (ViT/CNN) demonstrate that the VPFA framework is architecture-agnostic and robust. The clarified limitations and the discussion of semantic direction, as well as the plan for improved appendix visualizations enhance the paper's academic depth and clarity.
> >
> > Overall, the rebuttal successfully clarifies the scope and effectiveness of the work. I am satisfied with the proposed revisions.

---

> > > ### Author Response · Authors · 2026-04-03
> > >
> > > Thank you very much for your recognition of our work and for your valuable suggestions. We will carefully carry out the revisions and plans we have described in the next version, and we sincerely hope you will continue to follow our paper.

---

### Official Review · Reviewer_TPKp · 2026-03-11

**Soundness:** 3
**Presentation:** 2
**Significance:** 3
**Originality:** 3
**Overall Recommendation:** 3
**Confidence:** 4

**Summary:**

This paper studies cross-resolution person re-identification (CR-ReID), where a low-resolution (LR) query image must be matched to high-resolution (HR) gallery images, which is common in real surveillance footage. The authors argue that many existing methods either rely on super-resolution (extra compute/complexity) or try to learn resolution-invariant embeddings (which can mix up identity and resolution effects).
The main observation is that when you average out identity differences, the gap between HR and LR features extracted by common ReID backbones looks like a fairly consistent “shift” in feature space. The paper supports this with cosine-similarity tests on mean HR–LR difference vectors from different identity splits, plus correlation/CCA analyses.
Based on this, the paper proposes Vector Panning Feature Alignment (VPFA): after extracting features with a frozen ReID backbone, a small MLP predicts a correction vector for LR features. This correction is applied with a gated residual update to produce a pseudo-HR feature. Training uses an MSE regression loss between the corrected LR feature and the corresponding HR feature. The backbone is not retrained.
Experiments on several CR-ReID benchmarks (Market-1501, CUHK03, VIPeR, CAVIAR) under a multi-LR protocol show notable gains over a TransReID baseline, with additional efficiency analysis and qualitative visualizations.

**Compliance With Llm Reviewing Policy:**

Affirmed.

**Key Questions For Authors:**

1.	Training data construction: Do you train VP on per-image LR–HR feature pairs, or on identity-mean LR–HR features? The paper and implementation details seem to disagree. Also clarify what “identity-level shuffling” means.
2.	Fairness of comparisons: In the main table, are baselines re-implemented under the same backbone (e.g., TransReID) and the same multi-LR protocol, or copied from prior work using different settings? Please add a controlled comparison table.
3.	How general is the “resolution shift” effect? Does it hold across other backbones and other degradations (blur, compression artifacts, noise), not just bicubic downsampling?
4.	Does VPFA ever harm identity structure? Please report performance in HR→HR and LR→LR settings (and/or intra-/inter-class feature statistics) to show VPFA fixes resolution mismatch without collapsing identity separation.

**Limitations:**

At minimum, the paper should acknowledge:
•	VPFA depends on having LR–HR paired data for training (and in the described setup, identity labels).
•	It may be brittle if real LR degradations differ from synthetic downsampling.
•	Performance under extreme or camera-dependent resolution gaps is unclear.
•	Dual-use/privacy risks for surveillance applications, with concrete discussion and mitigation ideas.

**Strengths And Weaknesses:**

Strengths
•	Learning a small mapping in feature space is straightforward and avoids heavy super-resolution pipelines or retraining the backbone.
•	The gated residual update seems like a good way to avoid “over-correcting” and damaging identity information.
•	The paper provides multiple analyses suggesting the HR→LR difference is structured and consistent across identities.
•	The within-backbone ablation and visualizations (t-SNE / Grad-CAM) help explain what VPFA changes.
•	As a post-hoc module applied only to LR queries, it looks easy to deploy.
Weaknesses
•	Unclear / inconsistent training description: The paper description suggests per-image LR–HR pairing, but the implementation details mention identity-mean features and identity-level shuffling. These are not the same and affect how “supervision-free” the method really is.
•	“Supervision-free / ID-agnostic” claims look overstated: If identity labels are used to compute ID-wise means or filter IDs, then the method uses supervision.
•	SOTA comparison is hard to interpret: The comparison table mixes different backbones/training protocols. Without controlled re-runs under the same backbone and protocol, it’s hard to judge “state-of-the-art.”
•	Claim about a single direction could be strengthened: The results suggest structure, but they don’t clearly prove the shift is strictly 1D; it could be a low-rank subspace.
•	Weak limitations / impact discussion: Person ReID has clear privacy/surveillance risks, and the current impact statement is too dismissive.

---

> ### Author Rebuttal · Authors · 2026-03-31
>
> Thank you very much for your careful reading of our manuscript.
>
> ## To Weakness 1 & Questions 1 & Limitations 1:
>
> We sincerely apologize for this inconsistency. **To clarify, our training does not use identity labels, but image-level LR–HR pairs.** The implementation details **mistakenly retained wording from an earlier version of our method**, which used identity-mean features based on our CCA and Pearson analyses. After finding that image-level LR–HR pair training performed better, we revised the main description but unfortunately did not update that part accordingly. **We sincerely apologize for this oversight and hope for your understanding.** Our code will be fully open-sourced to support community reproduction.
>
> ## To Weakness 2:
>
> As clarified in Weakness 1, **our training is based on paired LR–HR images rather than identity labels**, making it ID-agnostic and consistent with our findings.
>
> ## To Weakness 3 & Questions 2:
>
> All results in Table 3 were obtained under standard ReID protocols. **Although the backbones differ, the test protocol is identical for each dataset. In addition, our cross-resolution data construction follows the widely adopted protocol in this field, ensuring fair comparison.** Therefore, the reported SOTA results are objective.
>
> ## To Weakness 4:
>
> We agree that the current evidence does not strictly prove a one-dimensional HR–LR shift, and we will state this more cautiously in the revision. Our CCA and Pearson analyses suggest shared structured variation rather than random noise, but do not exclude a low-rank subspace. In addition, our method uses an **MLP to learn an input-dependent nonlinear mapping, rather than a fixed linear shift**; Table 9 also shows that directly **adding a fixed shift vector performs worse than the full model**. **Therefore, our finding is better interpreted as a low-rank structured shift.** We will clarify this and discuss it as a limitation in the revised manuscript.
>
> ## To Weakness 5 & Limitations 4:
>
> Thank you for your reminder. We will add a limitations discussion and clarify the impact statement in the revision. **All ReID datasets used in this work are publicly available, legally accessible, and free of ethical controversy.**
>
> ## To Questions 3:
>
> This phenomenon generalizes across different backbones and training paradigms. **Table 6 shows the effectiveness of VPFA with the ResNet-based SAAI framework, while the supplementary results below show consistent gains with DeiT in TransReID and with both ViT and CNN in CLIP-ReID, further demonstrating its generalizability.**
>
> | Model | Ori (R1/R5/R10) | VPFA (R1/R5/R10) |
> |---|---:|---:|
> | TransReID (ViT)  | 90.3/96.1/97.5 | 94.1/97.6/98.6 |
> | TransReID (DeiT) | 89.1/95.7/97.1 | 91.7/96.5/97.7 |
> | CLIP-ReID (ViT)  | 90.5/96.0/97.2 | 93.4/97.3/98.6 |
> | CLIP-ReID (CNN)  | 88.2/95.6/97.1 | 90.6/96.1/97.4 |
>
> We agree that other degradations, such as blur, compression artifacts, and noise, are common in real-world ReID, but are not addressed in our current work. **Our method is mainly designed for resolution discrepancies, especially those caused by depth-of-field effects.** Thus, robustness to these degradations is a limitation of the current work, which we will clarify in the revision and explore in future work.
>
> ## To Questions 4:
>
> **During inference, HR features do not pass through VPFA, so HR→HR is identical to standard ReID.** Under the common CR-ReID protocol, only the query is downsampled. To further address your concern, we also **test the LR→LR setting** by downsampling all gallery images at inference. **On Market-1501, the Rank-1/5/10 results are 93.6/97.3/98.4, showing that VPFA remains effective without severely harming identity structure.**
>
> In addition, **the t-SNE results in Figure 3 qualitatively shows improved intra-class compactness with little change in inter-class structure.** We also **measure the Euclidean distances between HR and LR feature centers** for representative identities, and the distances are consistently reduced after VPFA, with an **average reduction of about 25%**.
>
> | ID | 1401 | 865 | 877 | 1355 | 507 |
> |---|---:|---:|---:|---:|---:|
> | Before | 17.87 | 22.53 | 15.32 | 15.26 | 15.15 |
> | After | 13.78 | 16.34 | 11.28 | 11.10 | 10.87 |
> | Reduction | 22.92% | 27.46% | 26.35% | 27.24% | 28.18% |
>
> ## To Limitations:
>
> Regarding the gap between real LR degradations and synthetic downsampling, **we follow the standard protocol for fair comparison and also validate our method on CAVIAR.** We agree that this is an important practical issue, and **will discuss it as a limitation** and evaluate it further on larger real-world datasets.
>
> Regarding extreme or camera-dependent resolution gaps, **our method is limited by the backbone’s feature extraction ability under very low resolution.** We will clarify this limitation in the revision. Although **current benchmarks partially cover camera-dependent gaps**, more challenging cases are still needed, which we will explore in future work.

---

> > ### Author Rebuttal · Reviewer_TPKp · 2026-04-03
> >
> > Thank you for the detailed rebuttal. The response resolves the inconsistency in the training description: I now understand that VPFA is trained on image-level LR–HR pairs, and that the identity-mean wording in the implementation details came from an earlier version. The clarification that the observed shift should be interpreted more cautiously as structured / potentially low-rank, rather than strictly one-dimensional, is also helpful. The additional results across more backbones and the extra LR->LR evidence strengthen the paper.
> >
> > However, my main concern is not fully resolved. In particular, the rebuttal does not yet provide the controlled comparison I asked for to support the main SOTA claim under matched backbone/training conditions. Stating that the protocol is standard and identical is helpful, but it does not remove the comparability issue when methods are mixed across different backbones and training pipelines. I also still think the limitations / impact discussion needs a more concrete treatment of privacy and surveillance risks, and the robustness claims should be more clearly scoped to the resolution discrepancy actually studied here.
> >
> > Because these remaining issues affect the interpretation of the main empirical claims and would require substantive revision rather than a short rebuttal, I am keeping my score unchanged.
> >
> > What still needs to be done:
> > fix the training-description inconsistency in the manuscript, either add controlled matched-setting comparisons or tone down the SOTA claim, frame the main finding as structured/possibly low-rank rather than strictly 1D, and substantially strengthen the limitations/ethics discussion around surveillance/privacy and robustness to real degradations.

---

> > > ### Author Response · Authors · 2026-04-04
> > >
> > > Thank you very much for your response to our rebuttal. We are glad that some of your concerns have been addressed. For the remaining issues you raised, we provide more detailed explanation and clarification below, and we sincerely hope to gain your further recognition.
> > >
> > > ## Explanation Regarding the Controlled Comparison Concern for the Main SOTA Claim
> > >
> > > **We would like to clarify that our SOTA claim is intended to be objective and fair.** All baselines in our comparison are evaluated under the commonly adopted benchmark construction protocols in the cross-resolution ReID literature, **using the same train/query/gallery splits for each dataset.** While we directly report the results published in prior papers, these methods are compared **under the same standard dataset settings and evaluation protocols**, which ensures fairness at the benchmark level. It is true that different CR-ReID methods may use different backbones, but this **is also the standard practice in this research area: the training sets and evaluation sets are shared, while the model design varies across methods, and such comparisons are an essential part of methodological progress in the field.**
> > >
> > > At the same time, we do not claim that TransReID provides any special advantage in this context. **TransReID is a widely used ReID framework from 2021 and is not uniquely stronger than other CR-ReID backbones by design.** To further support this point, we have also provided additional ablation and extension experiments.
> > >
> > > In addition, our method is a post-processing module. To verify its effectiveness, we chose a widely used standard ReID framework as the backbone. **Our intention is not to retrain or redesign a ReID model specifically for the cross-resolution setting, but to show that simply adding a lightweight post-processing module on top of a standard backbone can already surpass existing CR-ReID methods.** This is the key message we intended our experiments to convey, and we will further clarify this point in the revised manuscript.
> > >
> > > ## Explanation Regarding Privacy and Surveillance Risks
> > >
> > > We thank the reviewer for raising the concern regarding potential privacy risks. We acknowledge that person re-identification may be misused in surveillance contexts. However, **our work focuses on representation learning and identity consistency modeling under controlled academic settings, rather than real-world deployment for tracking individuals.**
> > >
> > > All experiments are conducted on publicly available benchmark datasets, **which are standard in the research community and do not contain explicit personally identifiable information such as names or contact details**. Besides, ReID research typically focuses on clothing appearance and body-level features rather than sensitive biometric identifiers such as facial information. Furthermore, our model operates on high-dimensional embedding features instead of explicit identity information, making it difficult to recover personally identifiable details.
> > >
> > > **We strongly oppose any misuse of this technology and emphasize that our work is intended solely for research purposes.** We will further explore privacy-preserving techniques such as feature anonymization and federated learning in future work.
> > >
> > > ## Explanation Regarding Robustness
> > >
> > > We state in the paper that the robustness of our method is supported by its effectiveness and potential in both cross-modal ReID and text-image ReID. **In addition, in Table 14, we further conduct cross-dataset evaluation by training on the training set of one dataset and testing on the test set of another dataset.** Our method still achieves strong performance under this setting, which further strengthens the generality of the proposed approach.
> > >
> > > We will further refine the discussion of robustness in the revised version of the paper to make it clearer and easier to understand.
> > >
> > > ## Explanation Regarding Substantive Revision
> > >
> > > We sincerely apologize that, at the rebuttal stage, we are unable to revise the main manuscript directly, even though we would be very willing to do so. **In the revision, we will explicitly add a clearer discussion of the limitations and ethical issues, and we will also correct the unclear statements in the current manuscript. These revisions are straightforward to make.** Thank you again for your valuable suggestions. We hope that the above clarification further addresses your concerns and earns your recognition.

---

### Official Review · Reviewer_3Ppn · 2026-03-17

**Soundness:** 3
**Presentation:** 2
**Significance:** 2
**Originality:** 3
**Overall Recommendation:** 4
**Confidence:** 4

**Summary:**

The paper addresses a challenging cross-resolution person re-identification problem. The paper reports novel experimental findings that, after averaging out identity-specific variation, the HR-LR feature discrepancy produced by standard ReID backbones exhibits a consistent, resolution-related semantic direction in the embedding space. The paper explores whether the feature space of CR-ReID contains analogous directions that encode resolution differences. Based on these findings, the paper introduces Vector Panning Feature Alignment (VPFA), a lightweight post-hoc module that learns to pan LR features along the learned resolution direction to obtain pseudo-HR representations. This proposed module aims to align low-resolution features with high-resolution features.

**Compliance With Llm Reviewing Policy:**

Affirmed.

**Final Justification:**

My Concerns are addressed. Based on the concerns addressed, please include these clarities and rebuttal results in the paper.

**Key Questions For Authors:**

- Refer to the Strengths and Weaknesses section.
- The paper should be clearly written with properly referenced Tables and a detailed analysis of the obtained results.

**Limitations:**

- The paper does not discuss the limitations of this work.
- The impact statement written in the paper is vague and not written clearly.

**Strengths And Weaknesses:**

**Strengths:**
- The proposed VPFA module leads to significant performance improvement on the TransREID model.
- The paper proposes an idea based on novel observation related to the resolution-related semantic direction in the embedding space, inspired by the semantic vector difference generally utilized in NLP tasks.
- VPFA is a lightweight post-hoc MLP-based module which can be integrated to existing ReID frameworks.


**Weakness:**
- To show the effectiveness of the VPFA module integrated as a post-hoc module to any existing ReID frameworks without retraining as suggested in the paper in Lines 362-364, please show results on multiple existing frameworks and integrate them with VPFA to show the performance improvement after VPFA. Currently, in the paper, it is just shown with one TransReID framework, which is not sufficient to show the effectiveness/generalizability of VPFA integration.
-  Please provide a proper justification for Table 1: If the downscaling factor is increasing from 2x to 4x, then why is the cosine similarity increasing, instead it should decrease as identity-related features in the downscaled images are less compared to the high-resolution image.
- What is the original size of the image, in order to identify the downscaled size of the image?
- What is the testing protocol used for the inference task? Does the proposed VPFA-based model evaluated on the sample testing protocol of the datasets, in order to ensure fair evaluation?
- The paper states that, “In the absence of semantic alignment, these correlations would cluster around zero”, but no reasoning and justification are provided for this statement.
- In Table 3, the results are mentioned on the TransReID model after integrating VPFA, including results without VPFA as well, especially on the real-world low-resolution CAVIAR dataset.
- The analysis and reasoning of results are limited. The paper only focuses on the outperforming numerical values reported in the paper, which is good, but proper reasoning should be provided for them. Similarly, in Table 2, how does the paper ensure that the obtained results are related to resolution variations observable at both global and identity-specific levels, rather than artefacts/noise induced in the image due to downscaling? Provide some reasoning for this, specifically relating it to the identity-specific levels.
- The paper is not well written. It has many non-referenced Tables in the sections: A.2, A.4, and A.5.
- The paper states that, “minimizing $L_{VPL}$ aligns both the direction via cos $\theta$ and magnitude of the features”, there must be a detailed discussion about aligning direction via cos $\theta$ and magnitude of the features.

---

> ### Author Rebuttal · Authors · 2026-03-30
>
> Thank you very much for your careful review. It has been very helpful to our paper, and we will try to address your questions one by one.
>
> ## To Weakness1:
>
> In Table 6 of our paper, we present the performance of VPFA on a cross-modality task, where the baseline SAAI is built on a ResNet-based framework that is entirely different from TransReID. **We have also added experimental results using DeiT within the TransReID framework, as well as results using both ViT and CNN within the CLIP-ReID framework**, which further demonstrate the effectiveness and generalizability of VPFA integration.
>
> | Model | Orignal (R1/R5/R10) | VPFA (R1/R5/R10) |
> |---|---:|---:|
> | TransReID (ViT)  | 90.3/96.1/97.5 | 94.1/97.6/98.6 |
> | TransReID (DeiT) | 89.1/95.7/97.1 | 91.7/96.5/97.7 |
> | CLIP-ReID (ViT)  | 90.5/96.0/97.2 | 93.4/97.3/98.6 |
> | CLIP-ReID (CNN)  | 88.2/95.6/97.1 | 90.6/96.1/97.4 |
>
> ## To Weakness 2:
>
> We are sorry that the unclear description in our paper may have caused your misunderstanding.**Table 1 reports the cosine similarity between two resolution-difference vectors, each computed from the difference between averaged HR and LR features.** Averaging is used to suppress identity-specific information and isolate the feature pattern caused by resolution change itself.
>
> ## To Weakness 3:
>
> In Market-1501, all images are resized to **64×128**. In CUHK, the image size is approximately **120×300**. In VIPeR, the average image size is **48×128**. CAVIAR is a real-world high-/low-resolution dataset, with image sizes ranging from **17×39 to 72×141**.
>
> ## To Weakness 4:
>
> **The evaluation protocol used for inference is consistent with the standard settings of the corresponding ReID datasets.** The construction of query and gallery in our dataset also strictly follows the common protocol of cross-resolution ReID. For datasets requiring repeated trials, we conducted the experiments accordingly, and the results are also provided in the appendix.
>
> ## To Weakness 5:
>
> Thank you for pointing this out. We agree that the original wording, “In the absence of semantic alignment, these correlations would cluster around zero,” was not sufficiently rigorous. **What we intended to convey is that, without a shared resolution-related direction, these correlations would not be consistently positive.** We will revise this statement to clarify that Pearson analysis is only supportive evidence and, together with CCA, suggests that the HR–LR discrepancy is structured rather than random.
>
> ## To Weakness 6:
>
> In Table 4 of the paper, we present the Market-1501 results with and without VPFA. Here we would also like to provide the corresponding results for CUHK, VIPeR, and CAVIAR, which will also be added to the main paper in the revised version, together with the details of repeated trials. Their Rank-1/5/10 results are **91.6/98.5/99.4, 46.5/73.7/82.2, and 60.6/81.8/92.4**, respectively. In all cases, VPFA consistently brings clear performance improvements.
>
> ## To Weakness 7 & Limitations 1:
>
> Thank you for this question. First, the multi-LR setting and downsampling strategy we adopted **follow the common evaluation protocol in the CR-ReID literature**; therefore, the experiments in Table 2 are based on downsampled images. Second, to ensure that the obtained results reflect resolution variations observable at both the global and identity-specific levels, we use averaging to reduce the influence of identity, and difference vectors to further suppress irrelevant noise, since random noise tends to be canceled out through averaging and differencing when a sufficient number of images are involved. That said, this is only our current practice for ensuring fair comparison while following the standard protocol of the field. **Your question has prompted us to think more carefully about this issue. We agree that this is a limitation of the current paper, and we will clarify it in the revised manuscript and further investigate its underlying nature in future work**.
>
> ## To Weakness 8 & Question 2:
>
> Thank you very much for your careful comments. We will correct these errors in the revision and further proofread the manuscript.
>
> ## To Weakness 9 :
>
> Referring to $L_{\mathrm{VPL}} = r^2 + R^2 - 2rR\cos\theta$, **minimizing this loss penalizes both angular error and norm mismatch, thus jointly aligning direction and magnitude.** We do not claim that VPL explicitly decouples these two factors; rather, as a Euclidean-distance objective, it implicitly constrains both. **This also helps explain its compatibility with downstream cosine matching.** We will clarify this in the revised manuscript.
>
> ## To Limitations:
>
> Thank you for your reminder. We apologize that the current manuscript does not discuss the limitations of this work. **In the revision, we will add a limitations section and clarify the impact statement.** We will also specify that all ReID datasets used in this work are publicly available, legally accessible, and not associated with ethical controversy.

---

> > ### Author Rebuttal · Reviewer_3Ppn · 2026-04-02
> >
> > Thank you for addressing most of my comments.
> >
> > However, I have a question regarding this: "Table 1 reports the cosine similarity between two resolution-difference vectors, each computed from the difference between averaged HR and LR features."
> > - Are the authors first computing the difference between HR and LR features and then computing the CCA between which two vectors? Can authors please elaborate in detail on the computation of CCA, including the formula they used and the vectors used for the computation?
> > - Further, do the authors want to convey through Table 1: High CCA value between very low-resolution (4x) and high resolution images leads to most semantic alignment, whereas low CCA value between low-resolution (2x) and high resolution images leads to least semantic alignment? Isn't it supposed to be reversed, because low-resolution (2x) images have more semantic information with respect to high-resolution images, as compared to very low-resolution (4x) images have less semantic information with respect o high resolution images. How are the authors relating correlation values with semantic alignment, please describe it technically.

---

> > > ### Author Response · Authors · 2026-04-03
> > >
> > > Thank you very much for carefully reviewing our rebuttal and for your further feedback. **In response to your new questions, we provide additional clarification below and will revise this part of the paper to make it clearer and easier to follow.** We sincerely hope this will address your concerns.
> > >
> > > ## To Questions 1 & the explanation of how the correlation values are related to semantic alignment:
> > >
> > > **For CCA (Table 2(a)), we do not analyze two "difference vectors".** Instead, we apply CCA to the LR and HR feature matrices to **measure their correlation in a shared global linear subspace.** In contrast, **Table 1 uses averaged HR–LR difference vectors as a discovery-oriented analysis** that motivated the subsequent CCA study.
> > >
> > > Our CCA experiment takes as input a set of LR feature matrices and their corresponding HR feature matrices. Let the low-resolution and high-resolution feature matrices be defined as
> > >
> > > $$ X_{LR} \in \mathbb{R}^{N \times d}, \qquad X_{HR} \in \mathbb{R}^{N \times d}, $$
> > >
> > > where 𝑁 denotes the total number of valid HR–LR paired samples, and 𝑑 denotes the feature dimension. Each paired sample can be written as
> > >
> > > $$ \left(z_{LR}^{(i)}, z_{HR}^{(i)}\right), \qquad i=1,\dots,N. $$
> > >
> > > Our goal is to analyze whether there exists a stable global linear correspondence between LR and HR features.
> > >
> > > **The core idea of CCA is to find one projection direction in the LR feature space and another projection direction in the HR feature space such that the resulting one-dimensional projected variables are maximally correlated.**
> > >
> > > Let the projected variables be
> > >
> > > $$ u = X_{LR} w_{LR}, \qquad v = X_{HR} w_{HR}, $$
> > >
> > > then the first canonical correlation coefficient is defined by the following optimization problem:
> > >
> > > $$ \rho_1 = \max_{w_{LR}\,w_{HR}} \frac{ w_{LR}^{\top}\Sigma_{LR,HR}w_{HR} }{ \sqrt{w_{LR}^{\top}\Sigma_{LR,LR}w_{LR}} \sqrt{w_{HR}^{\top}\Sigma_{HR,HR}w_{HR}} }, $$
> > >
> > > where:
> > > - $\Sigma_{LR,LR}$ is the covariance matrix of LR features;
> > > - $\Sigma_{HR,HR}$ is the covariance matrix of HR features;
> > > - $\Sigma_{LR,HR}$ is the cross-covariance matrix between LR and HR features.
> > >
> > > After obtaining the first pair of canonical variables, CCA continues to solve for subsequent pairs under the constraint of being orthogonal to the previous ones, yielding multiple canonical correlation coefficients:
> > >
> > > $$ \rho_1, \rho_2, \rho_3, \dots, \rho_k. $$
> > >
> > >
> > > The 𝑖-th pair of canonical variables can be written as
> > >
> > > $$ u_i = X_{LR} w_{LR}^{(i)}, \qquad v_i = X_{HR} w_{HR}^{(i)}, $$
> > >
> > > with the corresponding canonical correlation coefficient
> > >
> > > $$ \rho_i = \operatorname{corr}(u_i, v_i) = \frac{ \operatorname{Cov}(u_i, v_i) }{ \sqrt{\operatorname{Var}(u_i)\operatorname{Var}(v_i)} }. $$
> > >
> > > In our experiments, we retain the first three canonical correlation coefficients and denote them in the table as
> > >
> > > $$ R1 = \rho_1, \qquad R2 = \rho_2, \qquad R3 = \rho_3. $$
> > >
> > > These values measure how strongly the LR and HR feature matrices correspond in the most correlated shared linear subspaces. Higher values indicate stronger linear alignment along the corresponding canonical directions.
> > >
> > > To verify that this correlation is not accidental, **we also use a random-control group by shuffling the sample order of $X_{HR}$, which breaks the original one-to-one correspondence.** Since the true pairings yield much higher canonical correlations than the random-control group, this suggests that LR and HR features of the same image share a stable directional relationship rather than random noise. In addition, **based on standard thresholds, values above 0.4–0.5 are generally considered moderate to strong linear associations, further supporting the existence of a resolution-related direction in feature space.**
> > > ## To Questions 2:
> > >
> > > **In Table 1, "High CCA value between very low-resolution (4x) and high resolution images leads to most semantic alignment" is not what we intended to convey.** Table 1 computes **the cosine similarity** between **two resolution-difference vectors.** Specifically, we first split the data by identity into two disjoint subsets. Within each subset, we perform identity-wise averaging for the HR features and their corresponding LR features, and then average across identities to obtain
> > > $$
> > > V_{HR1},\ V_{LR1},\ V_{HR2},\ V_{LR2}.$$
> > >
> > > We then compute the cosine similarity between
> > >
> > > $$
> > > (V_{HR1}-V_{LR1})
> > > $$
> > >
> > > and
> > >
> > > $$
> > > (V_{HR2}-V_{LR2}).
> > > $$
> > >
> > > **The near-1 results in Table 1 indicate that, once identity differences are averaged out, the HR–LR discrepancy exhibits a highly consistent direction in feature space.** In other words, there exists a stable resolution-related semantic direction.
> > >
> > > Our notion of semantic alignment does not mean that LR and HR are identical at the pixel or fine-detail level. Instead, **it refers to whether their features still follow a consistent structural variation pattern in the embedding space, that is, whether their dominant directions of change remain regular and coupled.**

---

### Decision · Program_Chairs · 2026-04-30

**Decision:**

Accept (regular)

**Comment:**

The paper received mixed recommendations, with three reviewers supporting acceptance after rebuttal and one reviewer updating from weak reject to borderline. Reviewers recognized the novelty and practical value of the proposed lightweight feature-alignment method, and found the reported gains meaningful. After rebuttal, most concerns were sufficiently clarified, including the training setup, the interpretation of the resolution-related shift, and the effectiveness of the method across multiple backbones and related baselines. In particular, the added results on CLIP-ReID with a CNN backbone further support that the benefit is not limited to Transformer-based features. A remaining concern was the strength of the main comparison under unmatched backbones, together with the need for a stronger discussion of limitations, ethics, and robustness boundaries. AC agrees that these points should be addressed in revision, but does not view them as sufficient to outweigh the paper's merit and value to the community. AC therefore recommends acceptance. In the revision, the authors should incorporate the rebuttal clarifications, moderate claims where appropriate, strengthen the discussion of limitations and ethical considerations, indicate backbone families more clearly in the main comparison table, include the added CLIP-ReID (CNN) result, and, if feasible, add an extra ImageNet-pretrained ResNet-50 comparison to better support the main comparison table.